



# Passive Tracer Modelling at Super-Resolution with WRF-ARW to Assess Mass-Balance Schemes

Sepehr Fathi[1,3], Mark Gordon[2], and Yongsheng Chen[2]

[1]Physics and Astronomy Department, York University, Toronto, Canada
[2]Earth and Space Science and Engineering Department, York University, Toronto, Canada
[3]Air Quality Research Division, Environment and Climate Change Canada, Toronto, Canada

**Correspondence:** Sepehr Fathi (sfathi@yorku.ca, sepehr.fathi@ec.gc.ca) and Mark Gordon (mgordon@yorku.ca)

**Abstract.** Accurate "super-resolution" ($\Delta x < 250$ m) atmospheric modelling is useful for several different sectors (e.g., renewable energy, natural disaster prediction), and essential for numerous applications such as downscaling of weather and climate information to finer resolutions. It can also be used to interpret environmental observations during top-down retrieval campaigns by providing complementary data that closely correspond to real-world atmospheric pollution transport and dispersion

conditions. In top-down retrievals (e.g., aircraft-based), errors in estimates can arise from assumptions about atmospheric dispersion conditions, uncertainties in measurements, and data processing. As discussed in this work and in our companion paper (Fathi and Gordon, 2022), super-resolution numerical model simulations can be utilized to investigate these sources of uncertainty and optimize the retrievals. In order to conduct a thorough model-based study of the atmospheric dynamical processes that can affect top-down retrievals, model simulations at super-resolutions on the scale of measurement frequency are required:

sufficient to resolve the dynamical and turbulent processes at the scale at which measurements are conducted. Here, in the context of our modelling case studies with WRF, we demonstrate a series of best practices for improved (realistic) modelling of atmospheric pollutant dispersion at super-resolutions. These include careful considerations for grid quality over complex terrain, sub-grid TKE parameterization at the scale of large eddies, and ensuring local and global tracer mass-conservation.

For this work, super-resolution ($\Delta x \leq 50$ m, $\Delta t \leq 1$ s) model simulations with Large-Eddy-Simulation sub-grid scale pa-

rameterization were developed and implemented using WRF-ARW. The objective was to resolve small dynamical processes inclusive of spatio-temporal scales of high-speed (e.g., 100 m/s) airborne measurements. This was achieved by down-scaling of reanalysis data from 31.25 km to 50 m through multi-domain model nesting in the horizontal and grid-refining in the vertical. Further, WRF dynamical-solver source code was modified to simulate passive-tracer emissions within the finest resolution domain. Different meteorological case studies and several tracer emission sources were considered. Model-generated fields

were evaluated against observational data and also in terms of tracer mass-conservation. Results indicated model performance within 5% of observational data in terms of sea level pressure, temperature and humidity, and agreement within one standard deviation between modelled and observed wind fields. Model performance in terms of tracer mass conservation was within 2% to 5% of model input emissions.



## 1 Introduction

Generating model simulations of atmospheric processes at high spatial and temporal resolutions (super-resolution) have numerous applications including hybrid physical-model and machine-learning applications (Onishi et al., 2019), the dynamic downscaling of coarse resolution climate and weather information (Watson et al., 2020), and urban-climate feedback studies (Wu et al., 2021). Super-resolution modelling products ($\Delta x < 100$ m, $\Delta t < 1$ s) can also provide desirable information at the scale of measurements during top-down campaigns which can be analyzed in conjunction with measurement data to:

interpret observations, quantify uncertainty in the measurements, test the validity of assumptions in the employed top-down methodologies, and help fill the information gap in measurements. In the context of mobile platform (e.g., aircraft) top-down source emission rate estimations, numerical model simulations can be employed in various approaches. These include off-line applications, where a meteorological model (e.g., Weather Research and Forecasting - WRF) is used to replicate conditions during airborne and/or ground-based observations. The model generated meteorological fields are often used to drive a separate

Lagrangian tracer dispersion model (e.g., HYSPLIT) either forward in time to simulate tracer concentrations at observation times and locations, or for inverse method analysis (Cui et al., 2015; Lauvaux et al., 2016; Kia et al., 2022). Previous airborne studies have also used model generated wind fields and aircraft measured concentrations for flux calculations and mass-balance analysis (Karion et al., 2015). The emission and transport of passive tracers can be simulated in-line with meteorological fields within the same modelling platform such as the Eulerian WRF model, for source emission characterizations at the scale of

observations (Ahmadov et al., 2015; Barkley et al., 2017; Nahian et al., 2020). For these applications, model generated fields are analyzed as complementary information for characterizing emissions based on airborne observational data. For instance, Ražnjević et al. (2022) have employed large-eddy-simulation (LES) modelling driven by reanalysis data for interpreting field observations of $CH_4$. Further, model simulations of tracer transport and dispersion have been previously used for assessing the uncertainties/errors in top-down retrievals and optimizing the observational approach (Conley et al., 2017; Fathi, 2017;

Angevine et al., 2020; Fathi et al., 2021; Fathi, 2022). Numerical model simulations can also be used for simulating ground-based and/or airborne observations, where model generated fields are used as a proxy for measurement data (virtual sampling). For a robust model-based study of observational methods, model resolutions must be chosen to resolve the time and length scales of the measurements. For example, Gasch et al. (2020) simulated aircraft-based Doppler Lidar measurements of wind fields through LES modelling at 10 m resolution to investigate airborne lidar measurements for a lidar range length of 72 m.

Fathi et al. (2021), used a regional chemical transport model with physical and chemical process representations (GEM-MACH), and was successful in evaluating the application of the mass-balance technique in top-down retrievals using model simulated fields as a proxy for the real world environmental fields. However, the relatively coarse resolution (2.5 km, 2 min) of the employed model was insufficient for the investigation of aircraft-based retrievals through virtual airborne samplings within the model simulated 4D fields. In airborne campaigns, environmental observations (e.g. wind, temperature, tracer concentra-

tions) are made while flying downwind or around emission sources. These data are then processed through various retrieval algorithms to estimate source emission rates (Peischl et al., 2010; Ryoo et al., 2019; Gordon et al., 2015). An underlying assumption common among retrieval algorithms is the steady-state conditions during the sampling time of several hours (Alfieri



et al., 2010). Data collection during aircraft-based in-situ measurements are made through 3D space and over time, thus: (a) any point in space along the flight path is visited only once, and (b) spatially adjacent data points are collected at different (consecutive) times. By assuming stationarity (e.g., wind, emissions), the observational data are assumed to be representative of the average conditions during the sampling time. However, time-varying conditions (whether due to turbulence or weather trends) can reduce the representativeness of the sparsely collected environmental data. To study these effects through model simulations, the model resolutions should be chosen to resolve dynamical processes (turbulence) at the spatio-temporal scales at which aircraft in-situ measurements are made. For instance, to simulate (and evaluate) in-situ measurements at a flying/sampling speed of 100 m/s (e.g., Conley et al., 2017; Gordon et al., 2015), the model should be able to simulate (and output) atmospheric fields at length and time scales of $\Delta x \leq 100$ m and $\Delta t \leq 1$ s. Recent real-case LES-modelling studies have commonly referred to such resolutions ($\Delta x \leq 250$ m) as "**super-resolution**" (e.g., Wu et al., 2021; Onishi et al., 2019; Watson et al., 2020), herein we use the same terminology to describe our WRF model simulations.

The modelling requirements described above, motivated the development of super-resolution micro and LES scale atmospheric tracer transport model simulations, fine enough to resolve smaller-scale flow details and the effects of turbulence and changing stability in atmospheric mixing of tracer concentrations downwind of point and area sources of emission, enabling:

1. thorough dynamical evaluation of the application of the divergence theorem and the mass-balance technique in inferring source emission rates,

2. investigating the effects of flight pattern in aircraft-based top-down retrievals, utilizing model 4D output database,

3. exploring improved sampling approach through optimized flight design and multi-platform (in-situ, remote) sampling,

4. exploring improved data analysis, post-processing, and interpolation/extrapolation methods needed for flux calculations based on airborne observations.

In this study, we present a proof of concept for performing super-resolution model simulations of atmospheric tracer transport and dispersion using WRF with the ARW (Advanced Research WRF) dynamical solver core. The concepts that are explored here include (a) the realistic modelling of the atmospheric boundary layer at large-eddy-simulation scale over complex terrain, (b) the mass-conserved modelling of atmospheric dispersion and transport of passive tracers under the conditions described in (a), and (c) generating modelling products at spatio-temporal scale of airborne observations (aircraft-based in-situ and remote measurements), useful for evaluating the observational methods and providing recommendations for future studies. We evaluate the performance of our model simulations against historical observational data from ground-based monitoring stations and aircraft-based observations from the 2013 JOSM (Joint Canada-Alberta Implementation Plan on Oil Sands Monitoring) airborne campaign (JOSM, 2013). We further assess the performance of our simulations in terms of global (over the entire modelling domain) and local (sub-domain) mass-conservation, by conducting 4D mass-balance analysis.

We explore three different cases (dates and times) during August and September of 2013 over Canadian oil sands (Athabasca, Alberta). We use reanalysis data as initial and boundary conditions for our case studies. To achieve the desired micro and LES scale resolutions, we perform multi-domain nested simulations with LES parameterization for the finest domains. Further, we





modify the WRF source code (dynamical solver) to simulate the release of passive tracers from points and area sources within the finest model domain. The novel modelling approach with WRF presented in this work is comprised of: (1) dynamical down-scaling of reanalysis data from synoptic to LES resolution, (2) super-resolution model simulations through horizontal nesting and vertical grid refining, (3) LES sub-grid parameterization, and (4) passive tracer transport and dispersion simulations. To

our knowledge, the combination of these capabilities in WRF modelling has not been explored extensively in the past where reanalysis-driven super-resolution dispersion modelling under local mass-conservation condition is conducted. In this work we discuss a series of modelling best practices for such simulations in the context of our case studies, for improved modelling of atmospheric pollutant dispersion. The modelling approach in this work is geared towards the assessment of mass-balance methodologies, but throughout we discuss the general usefulness of super-resolution modelling for generating highly resolved

(spatial and temporal) pollutant dispersion forecasts and their potential application in measurement planning and interpreting the observations. The model output data from the super-resolution simulations in this work are also used for evaluating the accuracy of aircraft-based emission rate retrieval methodologies in our companion paper (Fathi and Gordon, 2022).

## 2 Methods

### 2.1 Case Studies

For this work we chose our case studies from the times and locations of three emission estimation flights during the JOSM 2013 campaign over the Athabasca oil sand region (Alberta, Canada). This choice was made to enable qualitative comparisons to observations. We considered the geographical location of an oil sands facility, Canadian Natural Resources Ltd. (CNRL). We configured our WRF model domain centred over the CNRL facility. For our WRF simulations, we considered model simulation times overlapping those of three JOSM 2013 box flights over CNRL (see Table 1). Box flight refers to closed shape

(e.g. rectangular, cylindrical) flight paths around the target emission source, where aircraft-based measurements are used to estimate source emission rates using the mass-balance technique (Gordon et al., 2015). Note that case 2 on 26 August 2013 was a "rejected" case in the actual campaign analysis due to unsuitable atmospheric conditions for aircraft-based retrievals (Fathi et al., 2021), but it is analyzed here as an assessment of the super-resolution model.

**Table 1.** Three case studies during late August and early September of 2013 over oil sands facility CNRL. Times and locations for the case studies where chosen from three JOSM 2013 box flights.

|  | Case 1 | Case 2 | Case 3 |
|---|---|---|---|
| Date | 20 Aug | 26 Aug | 2 Sep |
| Start Time (Local Time) | 10:30 | 13:43 | 11:43 |
| Start Time (UT) | 16:30 | 19:43 | 17:43 |
| Duration (hh:mm) | 02:10 | 01:52 | 01:45 |
| Model Simulation Time | 15UT - 19UT | 18UT - 21UT | 15UT - 19UT |



For each of these three cases, eleven tracer emission scenarios/sources were considered. Table 2 provides the spatial details
for the different emission sources, including seven elevated point sources (representing stack emissions), two small area sources
(representing surface mines), a large area source (representing the tailing pond west of CNRL) and a long multi-section line
source (over the approximate extent of the Horizon Highway south of CNRL). Table 2 lists geographical coordinates and tracer
release heights (stack top height) for all of the sources. Horizontal dimensions are also provided (in brackets) for the line and
area sources. The horizontal dimensions for each of the point sources are equal to those of one grid cell in the finest model
domain. Coordinates and heights for CNRL1-4 correspond to actual (real world) stacks in the CNRL facility. The hypothetical
source CNRL0 is co-located with CNRL1 and 4, with stack-top/release height at over 4 times higher than the tallest facility
stack (CNRL1), simulating the initial (assumed instantaneous) plume rise due to buoyancy. Figure 1 shows a map of the region
with the location and spatial extent of case study emission sources marked/labelled in red. The large area rectangular surface
source represents the tailings pond on the west of the CNRL complex. The multi-section line surface source represents the
Horizon Highway south of CNRL. Two small area sources labelled as Mine 1 and Mine 2 represent emissions from surface
mine excavation sites within the CNRL complex. Figure 1 also shows the locations for two hypothetical point (stack) sources
CNRLs (south) and CNRLw (west). During two of our case studies on 20 August 2013 (case 1) and 2 September 2013 (case
3), the mean wind was from west and south-west, placing these two hypothetical stacks upwind of the CNRL facility.

**Table 2.** Eleven tracer emission scenarios including seven point sources representing release from stack tops at various heights, three area
sources including a large area tailing pond towards the western side of the facility and two smaller (in area) surface mines, and a line source
approximately spanning the extent of the Horizon Highway south of the facility. Note that height/locations for sources with superscript ‡ are
hypothetical.

| Source ID | Type | Lat. | Lon. | Height agl (m) | Description |
|---|---|---|---|---|---|
| CNRL0[‡] | Point | 57.339 | -111.738 | 483 | Stack |
| CNRL1 | Point | 57.339 | -111.738 | 114 | Stack |
| CNRL2 | Point | 57.337 | -111.740 | 54 | Stack |
| CNRL3 | Point | 57.336 | -111.732 | 30 | Stack |
| CNRL4 | Point | 57.339 | -111.738 | 54 | Stack |
| CNRLw[‡] | Point | 57.327 | -112.014 | 102 | Stack (upwind west) |
| CNRLs[‡] | Point | 57.250 | -111.867 | 102 | Stack (upwind south) |
| HWY | Line | 57.258 | -111.765 (∼20 km ) | 6 | Horizon Highway |
| POND | Area | 57.348 | -111.918 (∼50 km$^2$) | 6 | Tailing Pond |
| MINE1 | Area | 57.337 | -111.834 (550 m × 550 m) | 6 | Surface Mine 1 |
| MINE1 | Area | 57.325 | -111.820 (350 m × 550 m) | 6 | Surface Mine 2 |



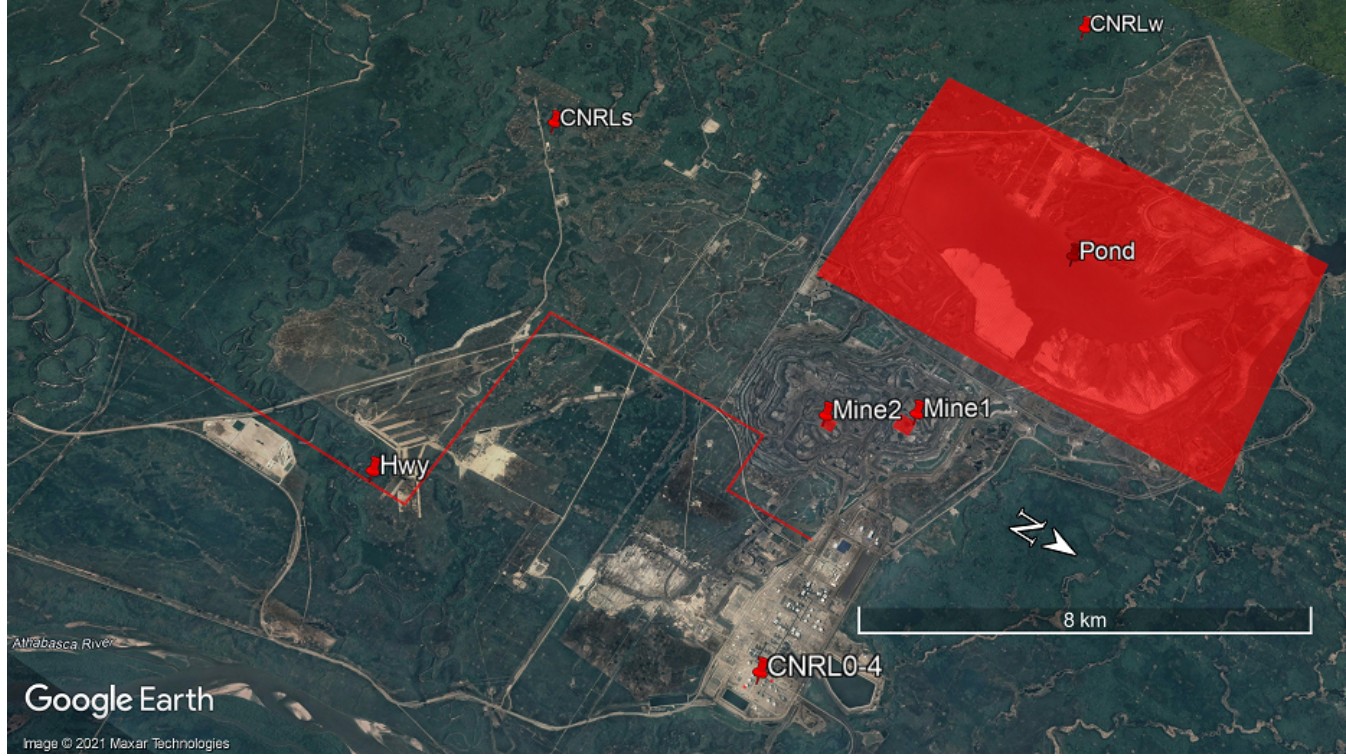

**Figure 1.** © Google map (2021) of the study region: The oil sands facility CNRL with case study emission sources marked on the map including seven point sources, two area sources, a large area surface source (tailing pond) and a multi-segment line source (road/Hwy). Direction of north (N) is shown with a compass arrow.

## 2.2 Model and Technical Setup

For this work, the Weather Research and Forecasting (WRF - Skamarock et al., 2008) model with the ARW dynamical core was utilized. WRF-ARW provides a multi-scale simulation framework suitable for efficient parallel computing with a vast range of physical parameterizations adaptable for different scales and dynamical processes. For this project, the High Performance Computing (HPC) resources of Compute Canada (CC) were used. WRF model simulations were run on 20 nodes with 20 cores and 128GB RAM memory per node on CC's super computers Graham and Cedar. To further speed up the model runs

and model output, multiple nodes were used for parallel I/O using the PNETCDF module embedded in the WRF modelling platform, which sped up the output part of the modelling process by 6-10 times compared to the conventional serial I/O. Model simulations generated 12GB of output data for each model second. The super-resolution WRF simulations for our case studies generated over 120TB of model output data, which were central to the purposes of this research work and are analyzed here and in our companion paper (Fathi and Gordon, 2022).

Mesh refinement and increased resolution can be achieved in WRF through series or concurrent grid nesting in the horizontal dimensions. With concurrent grid nesting, multiple computational domains with increasing resolution can be integrated



**Table 3.** Case study model setup for model simulations with 5 domains with increasing resolution. The first 3 domains have the same coarse vertical grid. Domains d04 and d05 have increasing resolution via vertical grid refinement. The finest domain (d05) has $\Delta z = 11.62$ m for the first 40 levels near the surface. $\Delta x$ and $\Delta t$ show the horizontal grid size and the model simulation time-step for each domain, respectively. $X$ and $Y$ indicate domain dimensions. $nx$, $ny$ and $nz$ are the number of computational grid points in each direction. With model $Z_{top}$ at 15.623 km (15.350 km agl) and $P_{top}$ (pressure at model top level) at 10 kPa.

| Domain | Vertical Grid | $\Delta x$(m) | $\Delta t$(s) | $X$(m) | $Y$(m) | $nx$ | $ny$ | $nz$ |
|--------|---------------|---------------|---------------|--------|--------|------|------|------|
| d01 | Coarse | 31250 | 100 | 6281250 | 6281250 | 201 | 201 | 30 |
| d02 | Coarse | 6250 | 20 | 3131250 | 3131250 | 501 | 501 | 30 |
| d03 | Coarse | 1250 | 4 | 751250 | 751250 | 601 | 601 | 30 |
| d04 | Fine | 250 | 0.8 | 175250 | 175250 | 701 | 701 | 48 |
| d05 | Fine | 50 | 0.16 | 50050 | 50050 | 1001 | 1001 | 82 |

simultaneously; where the coarse "parent" domain's output is interpolated to provide initial and lateral boundary conditions for the fine "child" domain, a process referred to as one-way nesting. Two-way nesting is achieved when information from the "child" domain is aggregated to write the overlapping regions of the "parent" domain. At high resolutions ($< 3$ km), mesh

refining in WRF via grid nesting only in the horizontal dimensions limits the control over the grid aspect ratio which can lead to poor grid quality and numerical errors. It has been shown that grid quality affects the accuracy of numerical solutions (Lee and Tsuei, 1992; You et al., 2006). A procedure permitting vertical nesting for one-way concurrent simulation is developed and described in Daniels et al. (2016), which allows high resolutions in the order of meters while grid quality is maintained. This procedure permits one-way concurrent grid nesting in both the horizontal and vertical and this is herein utilized for WRF

simulations with 5 domains (d01 - d05) with increasing resolutions from $\sim 31$ km to 50 m in the horizontal dimensions and up to near surface vertical resolution of $\sim 10$ m in the finest domain d05.

The North American Regional Reanalysis (NARR) GRIB data (at 3 hour intervals) from NOAA (National Oceanic and Atmospheric Administration) archives were used, for August and September of 2013 (period of 2013 JOSM campaign) over the Athabasca oil sands region (Alberta, Canada). Realistic WRF-ARW simulations were carried out with concurrent one-way

grid nesting in the horizontal dimensions with 5 domains (d01-d05) at a ratio of 1:5 and with mesh grid refinement in the vertical for the two smallest and finest domains (d04 and d05) by consecutively increasing the number of vertical levels near the surface. The finest domain (d05) has a horizontal grid size of $\Delta x = 50$ m over the entire domain, $\Delta t = 0.16$ s model simulation time-step and $\Delta z = 11.62$ m for the first 40 full grid levels near the surface; $\Delta x$ and $\Delta t$ configurations are set as such to ensure Courant-Friedrichs-Lewy (CFL) stability criterion, $\Delta t < \Delta x/|u_{max}|$ where $|u_{max}|$ is the maximum wind speed

in the model (Jacobson, 2005). Table 3 provides the details of model grid configurations for the five domains.

Figure 2a shows the 5 domains of the model and their relative size. Model domains are centred on the region of Athabasca oil sands with the two finest domains (d04 and d05) centred on the CNRL Horizon facility on the north west quarter of the complex, west of the Athabasca river. Figure 2b shows a map of the region with the CNRL facility marked on the map (red



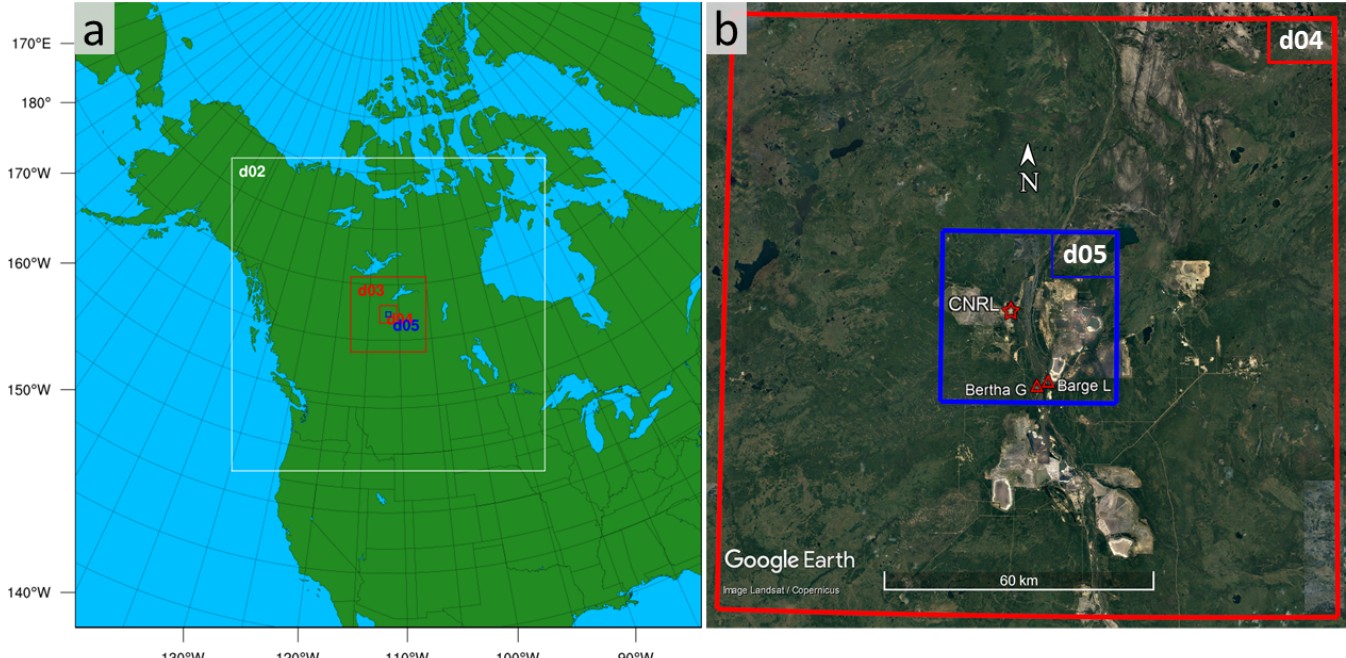

**Figure 2.** (**a**) WRF model grid (horizontal) configuration with five nests d01 - d05 with increasing resolution (decreasing size) at a ratio of 1 to 5. The finest domain, d05, is centred over the oil sands region. (**b**) © Google map (2021) of the oil sands region with overlay of model domains d04 and d05. The CNRL facility is marked with a red star. Locations for two WBEA monitoring stations are also shown with red triangles, Bertha Ganter – Fort McKay and Barge Landing monitoring stations.

star). Boundaries of the two finest domains, d04 and d05, are also overlaid on the map in Figure 2b. The relative location of
the other oil sands facilities can be seen in the figure. CNRL is at the north west corner of the complex with no facilities to its
north and west. The oil sands region is located on the Athabasca river valley with 400-500 m vertical relief within a few tens of
kilometres of the facilities (mainly in the west-east direction). This may give rise to complex flow and frequent vertical wind
shear in the valley (Gordon et al., 2018). In the following sections we discuss the performance of our super-resolution model
simulations against observed meteorology for the same locations and time periods.
Domain d05 has a vertical resolution of about $\Delta z = 12$ m for the first 40 levels near the surface and the vertical resolution
decreases gradually with altitude up to about 1000 metres above sea level (masl), beyond which it matches the vertical grid
resolution of domain d04. Vertical resolution of about 12 m for the bottom 500 m agl (above ground level) is sufficient for
investigating and evaluating different methods for extrapolating sampled data below the lowest flight level (typically $\sim 150$ m)
where no aircraft-measurements are usually made, which is required for flux estimations in top-down emission rate retrieval
methods (Gordon et al., 2015).



In order to simulate small-scale atmospheric dynamical processes, the finest two model domains (d04 and d05) were configured with the following Large Eddy Simulation (LES) sub-grid parameterization available in the WRF model (Skamarock et al., 2008),

1. Diffusion option was set to "Full Diffusion" mode (*diff_opt*= 2) to accurately compute horizontal gradients using full
metric terms.

2. "K Option" was set to *km_opt*= 2 to solve a prognostic equation for turbulent kinetic energy (TKE) where the diffusion coefficient K is calculated based on TKE (see Appendix A).

3. 6th Order Horizontal Diffusion was set to default option (*diff_6th_opt*= 1).

WRF 3.7+ includes a basic framework for initiating and creating continuous (and variable) emission tracer plumes. To create
tracer transport simulations, WRF dynamical solver FORTRAN source codes were modified following Blaylock et al. (2017). Tracer amounts were initiated at several different horizontal locations within the finest domain (d05) after 30 minutes model simulation. A meteorological spin-up time of 1 hour was considered through testing with the modelling setup for different initial and boundary conditions according to the following criteria: for the cases we considered, within the first hour of simulation time (a) the model boundary conditions propagated over the entire span of domain d05, (b) model winds (in west-east and
south-north directions) assumed continuous profiles both horizontally and between model vertical layers, (c) water vapour on model mass-layers (horizontal and vertical mass grid points) assumed continuous profiles within this time period. Tracer release for our considered emission scenarios (see Table 2) started after a 30 min spin-up time (half the meteorological spin-up time) and continued for the rest of the simulation period. These included surface emissions on several model grid points at the lowest level (i.e., level 1 at $\sim$ 6 m agl) at various locations, and stack emissions at levels 3, 5, 9, 10 and 40 according to the stack
top heights and horizontal locations described in Table 2. Note that in WRF-ARW vertical levels (Fig. S1) are configured using a terrain-following hydrostatic-pressure coordinate system (Skamarock et al., 2008) and therefore stack-top heights for our simulations are assigned to pressure levels with heights (in meters) closest to source heights. Depending on pressure changes, the height of pressure levels can vary over time. This variation was determined to be smaller than the corresponding vertical level thickness for our simulations and therefore its impact on tracer release simulation is considered negligible in this work.
All of the tracer emissions were implemented within the boundaries of the CNRL facility and the surrounding region, on the eastern half of the modelling domain. All the emission scenarios involved the emission of passive and non-buoyant tracers with no interactions with meteorology and no defined surface deposition rates. Note that the topography input information (land use indices) used in our WRF simulations were not modified to represent oil sands operations (e.g., tailing pond, excavation sites) and only represent natural features (e.g., river basin, hills). The release, dispersion and transport of tracers from our emission
scenarios under different meteorological conditions are discussed in the Results section.

## 2.3 Divergence Theorem and the Mass-balance Technique

For this work, the mass-balance technique is utilized for calculating the net integrated flux out of virtual control volumes (emission box) enclosing the emission sources of interest. The calculation steps in this section follow those in Fathi et al.





(2021), with slight modifications for use with WRF model output data. For a detailed discussion on the application of the

mass-balance and divergence theorem in estimating source emission rates see Fathi et al. (2021).

In applying the mass-balance technique to estimate the rate of emissions from sources within a flux box (control volume), the mass flux exiting the box through box top and lateral walls are equated to the emission rate of the tracer within the box. The processes contributing to the change of mass, for a passive tracer, within the control volume can be described with the following expression,

$$S_C = E_C - F_{C,H} - F_{C,V} - F_{C,HT} - F_{C,VT} \tag{1}$$

where the storage term $S_C$ represents the change in mass of tracer $C$ within the control volume, $E_C$ represents the tracer emission rate, $F_{C,H}$ and $F_{C,V}$ represent the net horizontal and vertical advective fluxes exiting through lateral and top walls of the flux box, respectively. $F_{C,HT}$ and $F_{C,VT}$ represent horizontal and vertical turbulent fluxes across the box walls, respectively.

The total mass of the tracer within the control volume can be calculated by integrating the mass over the entire volume of

the box at each model output timestamp ($\Delta t = 1$ sec),

$$B_{C,Tot}(t) = \iiint \chi_C(t,x,y,z)dxdydz \tag{2}$$

where $\chi_C(t,x,y,z)$ is the tracer concentration at each model grid point. Further the storage term can be calculated by taking the time derivative of $B_{C,Tot}(t)$,

$$S_C(t) = \frac{\partial}{\partial t}B_{C,Tot}(t) \tag{3}$$

Horizontal advective flux through the lateral walls of the box can be calculated by extracting tracer concentration and normal wind (positive outwards) along the lateral walls of the box from model output,

$$F_{C,H}(t) = \iint \chi_C(t,s,z)U_\perp(t,s,z)ds(x,y)dz \tag{4}$$

where $ds(x,y)$ is the path $s(x,y)$ increment along the walls. $U_\perp$ is the normal wind to the box walls (positive outwards).

Similarly, the vertical flux through the box top can be calculated as,

$$F_{C,V}(t) = \iint \chi_{C,top}(t,x,y)W_{top}(t,x,y)dxdy \tag{5}$$

where $\chi_{C,top}(t,x,y)$ and $W_{top}(t,x,y)$ are tracer concentration and vertical wind speed at box top, respectively. See Appendix B for turbulent flux terms. As we show later, the vertical advective ($F_{C,V}$) and the turbulent fluxes ($F_{C,HT}$, $F_{C,VT}$) have negligible relative contributions to the mass-balance equation, with the horizontal advective flux $F_{C,H}$ being the dominant term removing mass from the box. We collect all the flux terms (advective and turbulent) contributing to the removal of tracer

mass from the box into a single flux out term as $F_{C,out} = F_{C,H} + F_{C,V} + F_{C,HT} + F_{C,VT}$. By rearranging Eq. 1, the tracer emission rate can be estimated based on the other terms as,

$$E_C = S_C + F_{C,out} \tag{6}$$





By extracting the required fields from the model output 4D database, Equation 6 can be utilized to determine the source emission rate based on the mass-balance equation which can then be compared to the known input emission rate. Following 240 the above described calculation process, source emission rates can be estimated at each model output time-step and compared to the model input emissions to evaluate model performance in terms of local mass-conservation and mass-flux consistency.

## 3 Results and Discussions

Model simulations were carried out for the period between 15UT to 19UT for cases 1 and 3 on 20 August and 2 September 2013, respectively. The simulation period for case 2 on 26 August 2013 was between 18UT to 21UT (see Table 1). NARR 245 reanalysis data at 31.25 km resolution was used as initial and boundary conditions (IC and BC) for the coarsest domain d01 with the same resolution. Through model nesting at an increasing resolution (and decreasing size) ratio of 1:5, the input reanalysis data were down-scaled to consecutively higher resolutions all the way to 50 m in the finest domain d05. Each parent domain provided initial and boundary conditions for their nested child domain: d01 ⇒ d02 ⇒ d03 ⇒ d04 ⇒ d05. Note that feedback (two-way nesting) between parent and nested domain was turned off to allow for vertical grid refining for domains 250 d04 and d05. Output frequency was set to 3 hours for domains d01 – d03, 100 seconds for d04, and 1 second for d05.

### 3.1 Meteorological Evaluation

We compared output from domains d03, d04 and d05 for case 1 and found that output from the three finest domains agree to a great extent for sea level pressure with < 0.1% difference, relative humidity (at 2 m agl) with 2% difference, and temperature (at 2 m agl) with 0.1% difference. Wind directions were also consistent with 2 to 7 degrees difference for a mean wind direction 255 from about 240 degrees (west-south-west). Wind speeds were biased high relative to domain d03 with a mean wind speed of about 6 m/s, by 0.7 m/s to 1.5 m/s for domain d04 and by 3 m/s to 4 m/s for domain d05.

Output from the finest three domains were compared to the concurrent historical observational data from the Wood Buffalo Environmental Association (WBEA) continuous monitoring stations Bertha Ganter – Fort McKay and Barge Landing (https: //wbea.org/historical-monitoring-data/). Figure 2b shows the locations for the two WBEA stations on the map of the region 260 (red triangles). Both stations provide complete observational historical data for our periods of interest in August and September 2013. We compared model sea level pressure, 2-m relative humidity (Rh), 2-m temperature and 10-m wind to the corresponding WBEA observational data. Model values at 2 m and 10 m agl were determined by interpolating between model grid point values at the surface and at the top of the first model layer at ∼ 12 m agl. Figure 3 shows wind-rose diagrams for case 1 on August 20 where output from model domains d03, d04 and d05 are compared to WBEA data at the location of Bertha Ganter–Fort 265 McKay monitoring station. Wind directions for this case were from west and west-south-west during the simulation time, which is consistent with the observed WBEA wind directions. Model wind directions are within 20 to 30 degrees of the WBEA observational data. Model wind speeds are higher for all three domains compared to WBEA observational data for the locations of the two monitoring stations (with mean wind speed of 3 m/s): by 2 – 3 m/s for domain d03, by 3 – 4 m/s for domain d04, and by 4 – 7 m/s for domain d05. Note that the Large Eddy Simulation (LES) subgrid parameterization was used





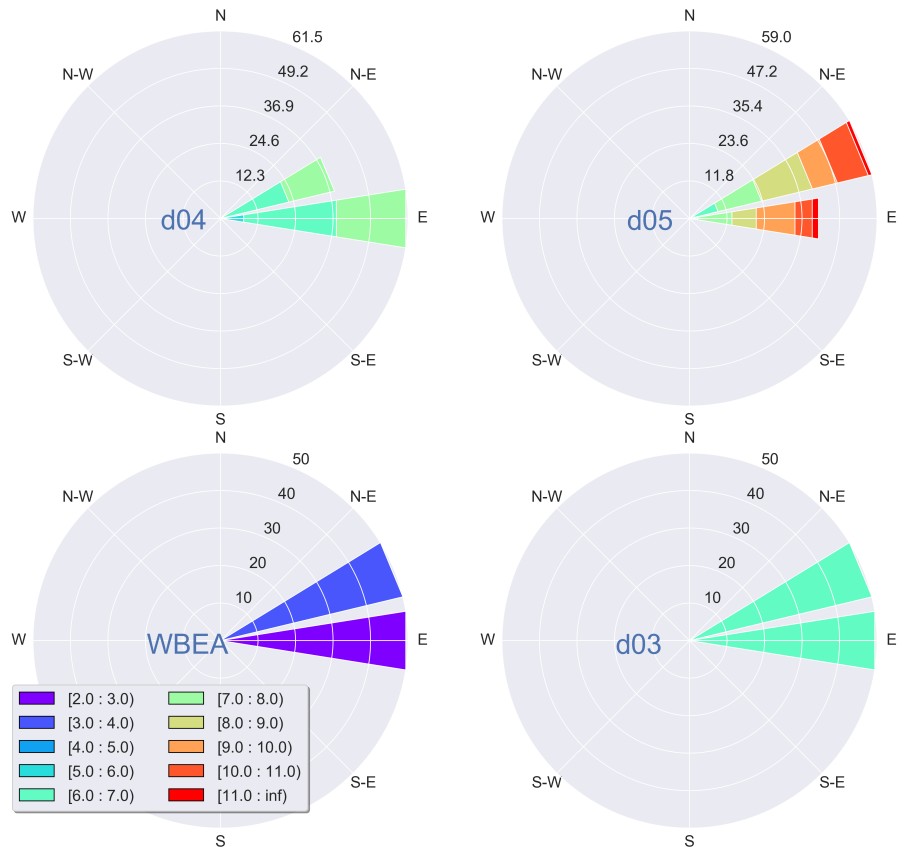

**Figure 3.** Wind-rose diagrams comparing observational data from WBEA monitoring station Bertha Ganter–Fort McKay to model output (case 1) winds at the location of the WBEA station from domains d03, d04 and d05 for 20 August 2013. Wind distributions, indicated on each circle, are in units of percentage. Wind directions are consistent within 20 degrees, blowing from WSW. Wind speeds are biased high at various degrees for the three domains.

270   for d04 and d05 simulations. Model winds for these two domains, especially for d05, are highly variable over time and space. Figure 4 shows output from d04 and d05 compared to observational data from the two WBEA monitoring stations for wind, 2-m temperature and 2-m relative humidity for case 1. Comparisons for the other cases show similar results, with agreements within 1% to 5% for 2-m temperature, 2-m relative humidity, sea level pressure and wind direction. Similar to case 1, wind speeds were higher than WBEA winds by between 2 m/s to 8 m/s for cases 2 and 3.

275   We note the following considerations when comparing model generated fields (e.g., wind) to WBEA observational data:

1. The lack of observational data from continuous monitoring stations for more spatial locations, especially closer to the centre of domain d05 (where CNRL is located) is a source of uncertainty. Note that the two available WBEA stations





**Figure 4.** Case 1 model output for 2-m temperature, 2-m relative humidity and 10-m wind speed from domains d04 and d05 are compared to observational data from WBEA monitoring stations.

are located close to the southern boundary of domain d05, less than 200 model grid points from the boundary, as shown in Figure 2b. Model fields close to domain boundaries are highly impacted by the boundary conditions from the parent domain, and are usually not included in model output analysis. For this work, we considered a buffer zone of 100 grid points on each side and excluded data from this zone in our analysis. We note that discrepancies between model fields and WBEA observational data are smaller for domains d03 and d04 compared to d05, where WBEA locations are well within the interior of the modelling domains (far from lateral boundaries).

2. The wind speeds in the NARR reanalysis data (at 31 km resolution) used as input for our simulations, were higher than WBEA observed values by 2-3 m/s for the region and the periods of interest. These are very similar to domain d03 output fields, which indicate the high fidelity in d03 domain simulations. As a result we can use d03 output fields at 1250 m





resolution as a base case for evaluating d04 and d05 simulations. Note that wind speeds were biased high by only about 1 m/s for d04 and by about 3 m/s for d05 simulations (see Fig S2 for evaluations against d03).

3. Dynamical down-scaling of NARR reanalysis data from 31.25 km resolution to 50 m resolution with five nested domains and vertical grid refining, is another source of uncertainty. In concurrent grid nesting as used in this work, output from parent domain is interpolated to provide initial and boundary conditions for each respective nested domain. Horizontal, vertical and temporal interpolation errors are therefore compounded with each nesting (five in this case). Daniels et al. (2016) demonstrated how horizontal grid nesting and vertical grid refining can result in 1-2 m/s bias in wind speeds for each nesting. This is consistent with our results where d04 wind speeds are higher than d03 by about 1 m/s, and d05 winds are higher than d04 by about 1-2 m/s.

4. The accuracy of subgrid scale LES parameterization in domains d04 and d05, is also a source of uncertainty. Liu et al. (2011) discussed concurrent nested modelling from synoptic scale to the LES scale with 4 domains. They demonstrate how simulated wind speeds differ by 2-5 m/s for the 4 domains, with weakest winds in the coarsest domain and stronger winds in the finest domain. Which is consistent with our results were wind speeds are biased high for the finest domain d05 compared to d04 and d03 by 1-4 m/s.

We also compared wind fields from domain d05 to aircraft observations during the 2013 JOSM campaign over the oil sands region for the same time periods as our model simulations. Figure 5 compares model wind speeds and directions for our three cases to aircraft observations for altitude levels of airborne measurements. WBEA data at 10 m agl are also included on the figure for comparison. Horizontal bars on Fig. 5 show one standard deviation in model-generated fields over domain d05. Model wind fields (vertical profiles) overlap with aircraft observation within one standard deviation for the three cases. WBEA wind speeds are lower than both model and aircraft wind speeds. Note the high spatial heterogeneity in wind fields captured by both aircraft observations ( Fig. 5, blue dots) and model simulations ( Fig. 5, orange bars). Spatial (horizontal and vertical) variability in wind fields were more severe for case 2 compared to the other two cases. We discuss later how the conditions of case 2 resulted in weak advection of tracer mass and rendered this case unsuitable for top-down mass-balance retrievals.

For all cases, output from domain d03 showed the best agreement to observational data. Since observational data were not available at the location of CNRL oil sands facility and due to the fact d03 output fields showed good agreement (see above) to observational data for the two monitoring stations, we evaluated the performance of d04 and d05 simulations to d03 as a proxy for observational data at the geographical location of CNRL. The evaluations were made in terms of east ($U$) and north ($V$) wind components, 2-m temperature, 2-m relative humidity and sea level pressure (see Figure S2 for case 1). Root mean square (rms) error scores were small (e.g., 0.5 m/s for $U$, 0.68 m/s for $V$, 0.45 °C for temperature) for d04 simulations at 250 m horizontal resolution and with LES parameterization. rms errors for d05 simulations were also similarly low for sea level pressure, temperature and humidity, with wind speed biased high by 1.57 m/s and 3.47 m/s for $V$ and $U$ respectively (at 10 m agl).

As the model output data from domain d03 were only for every 3 hours, to evaluate the performance of domain d05 simulations at CNRL at a higher temporal resolutions we performed evaluations against domain d04 with model output every





**Figure 5.** Model-generated wind fields (orange) compared to aircraft observations (blue dots) at several altitude levels during the 2013 JOSM airborne campaign, and WBEA observational data at 10 m agl (red diamonds). Horizontal bars show one standard deviation in model wind fields. Aircraft data and model fields agree within one standard deviation. WBEA wind speeds are lower than both aircraft and model wind speeds, red bars show one standard deviation in WBEA data.

100 seconds. Evaluation results for all three cases are summarized in Table 4. Root mean square (rms) error ranges from 0.09



**Table 4.** Evaluation of domain d05 simulations against d04 output fields at every 100 seconds at the geographical location of the CNRL oil sands facility (Lon=-111.738 and Lat=57.339). Note that positive/negative sings mean over/under-estimates by d05 relative to d04.

|  | Score | SLP (hPa) | 2m RH (%) | 2m T (°C) | 10m U (m/s) | 10m V (m/s) |
|---|---|---|---|---|---|---|
|  | d04 mean | 1006.42 | 51.18 | 19.32 | 5.05 | 3.20 |
| Case 1 | rms error | 0.11 | 1.89 | 0.25 | 3.16 | 1.37 |
|  | mean error | -0.04 | -1.20 | -0.17 | 3.11 | 1.14 |
|  | d04 mean | 1013.44 | 52.59 | 20.30 | -0.766 | 0.799 |
| Case 2 | rms error | 0.56 | 7.59 | 1.84 | 0.59 | 2.20 |
|  | mean error | 0.55 | 6.86 | -1.81 | -0.55 | -2.16 |
|  | d04 mean | 1006.14 | 62.40 | 21.95 | 3.57 | 3.90 |
| Case 3 | rms error | 0.09 | 2.44 | 0.38 | 2.77 | 1.01 |
|  | mean error | 0.06 | -2.24 | 0.36 | 2.73 | 0.75 |

hPa to 0.56 hPa for sea level pressure (SLP), 1.89 % to 7.59 % for 2-m relative humidity (RH), 0.25 °C to 1.84 °C for 2-m temperature, 0.59 m/s to 3.16 m/s for west-east wind component $U$, and 0.75 m/s to 2.20 m/s for south-north wind component.

For case 1 on 20 August 2013, 2-m temperatures ranged from 17 °C to 20 °C and 10-m wind speed from 6.6 m/s to 11.4 m/s at the geographical location of main CNRL stack sources. For case 2 on 26 August 2013, the 2-m temperature range was similar to case 1 but the 10-m wind speeds were much lower ranging from 0.8 m/s to 3.5 m/s with mean 2.8 m/s. For case 3 on 2 September 2013, 10-m wind speed ranged from 5.3 m/s to 9.4 m/s with 2-m temperatures slightly higher than the other cases ranging from 21 °C to 23 °C. The temperature and wind speed ranges mentioned for the three cases are for periods between 18:00UT to 20:00UT (local noon to 2pm).

## 3.2 Plume Characteristics

In this section we briefly discuss plume behaviour for our three case studies with emphasis on case 1 as our main case. Continuous passive tracer emissions for surface and stack sources were initiated simultaneously at different locations within the finest resolution modelling domain (d05), after a 30 min initial model spin-up time. Source locations, spatial extent, and release heights are provided in Table 2. Figure 6 shows tracer plumes for case 1 on 20 Aug 2013 at 17:35 UT, 2 hours and 5 minutes after the initial release. For the example shown, tracer plumes have propagated the downwind span of the modelling domain and reached the opposite lateral boundary. The mean direction of wind for the first 1.5 hour of the simulation for this case was from south west, which transitioned to winds from west and west-south-west in the following hours (Fig. 6). As mentioned before, we have discarded 100 grid points from the lateral boundaries of the domain and considered output data for the inner sub-domain in our analysis. The mass balance calculations discussed below start at 16:30UT (30 min after start up), as the addition of tracer mass through source emissions and removal via advective and turbulent fluxes through the boundaries of the control volume (box) have reached a mass-balance and a relative steady-state by this point in the simulation time.



**Figure 6.** Top view of the modelling domain (d05) and the location of tracer emission sources are shown. Sources include a big area source (Pond), a multi-segment line source (Highway), two small area sources (Mines 1, 2), five CNRL stacks 0-4 and two upwind point sources (west and south). Emission plumes from Pond are shown in blue, Highway in gray, Mines in green and stack/point sources in red. Colour darkness is proportional to log of column total tracer mass. Two control volumes (dark dashed boxes) for 4D mass-balance calculations are shown, one enclosing all the emission sources and the other marking the boundaries of the CNRL facility (the smaller box) and only including within facility sources.

Tracer amounts were advected at different vertical levels and flow regimes. Flow was different at other vertical levels. For instance, surface emissions from MINE1 and MINE2 (Fig. S3 top right panel) were advected in slower air flows and covered less downwind range during the same time period compared to stack (elevated) emissions CNRL0-4 (Fig. S3 top left panel).





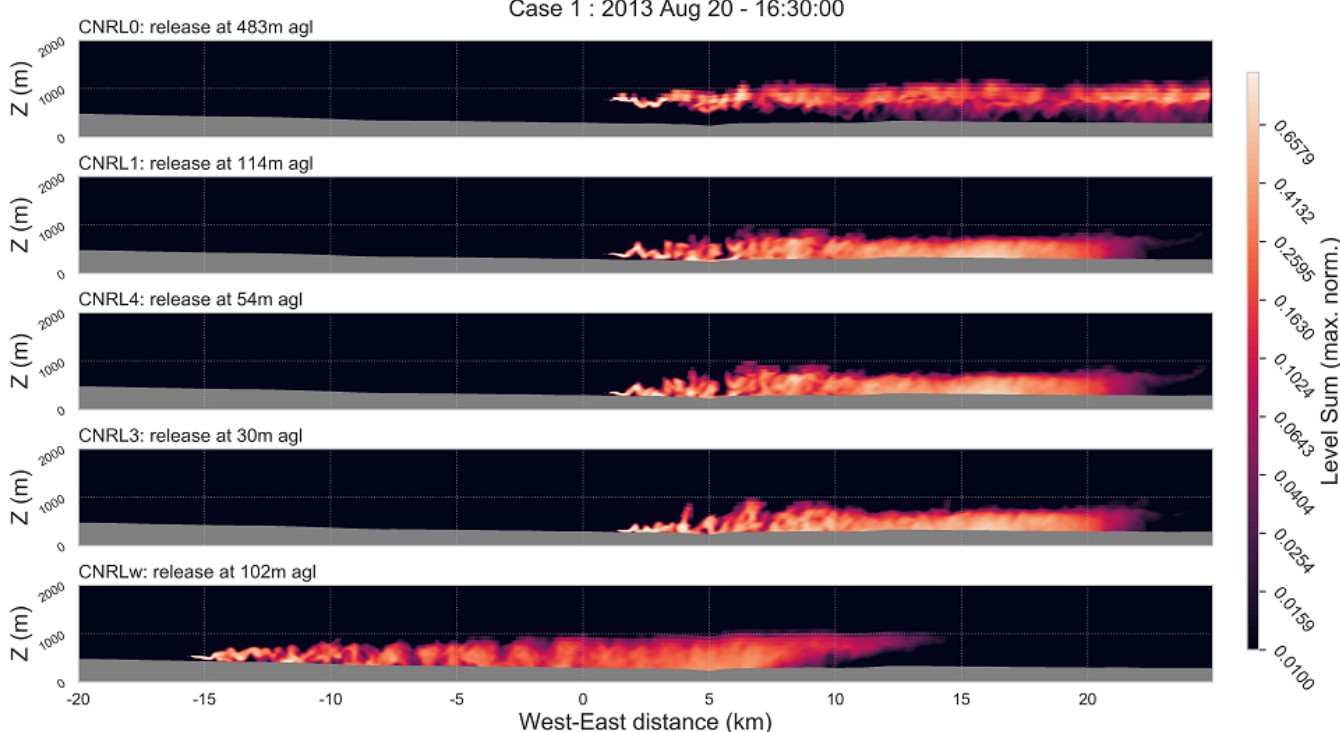

**Figure 7.** Domain d05 west-east vertical cross-section for case 1 on 20 August 2013. Vertical cross-section of tracer plumes from stack/point emission sources are shown. Release height above ground is indicated for each source. Data shown is the level tracer count sum normalized to maximum values for each source. The origin for west-east distance in km is at domain centre.

Figure 7 shows a west-east vertical cross-section of the modelling domain, with tracer plumes from select stack/point emission sources. Advection in west-east orientation is unidirectional at all vertical layers as can be seem in Fig. 7. The plume centre-line for CNRL0 emissions remains near the initial release height, while mixing in the vertical as it is advected downwind. CNRL1-4 emissions, although released at different heights above ground (by a few tens of meters), show similar vertical mixing profiles along the downwind advection path. Emissions from CNRL1-4, mixed to the ground surface within the 5 km

range and assumed a near uniform vertical mixing beyond 15 km downwind distance. Note how plumes from these sources interact with the Athabasca river basin at the 5 km distance. There is an apparent discontinuity in tracer concentrations beyond which mixing in the vertical is intensified. This is likely due to different surface fluxes over the river and the ground on either side with stronger updrafts. This interaction is less visible in CNRLw plume which is relatively well mixed by 5 km distance, but it can be seen that the vertical mixing becomes more uniform on the other side of the river for this plume.

Wind and therefore transport in the south-north direction were weaker than the the transport in the west-east direction. There was also a strong vertical shear in south-north wind ($V$). See Appendix Figure C1 for meteorological vertical profiles for this case. While all tracer plumes for release heights of 30 m to 114 m agl were advected north, CNRL0 at 483 m agl was




advected south (Fig. S4 top panel). Similar atmospheric processes governed the dispersion and transport of tracer plumes from the surface emission sources (see Figure S5).

WRF model simulations for case 2 for the period during 26 August 2013 started at 18UT. As mentioned before, the JOSM emission flight for this day was rejected for emission rate calculations in relevant publications due to low and variable wind speeds (Fathi et al., 2021). This case is referred to as "rejected" throughout this work. We recreated the meteorological and tracer transport conditions on this day with our super-resolution simulations, which are presented here as an example of unsuitable conditions for top-down retrieval. For this case, tracer emissions from various sources were initiated at 18:30UT. During the

period between 19UT to 21UT, wind fields over the region of interest were highly variable (spatially) with very low wind speeds (Figure S6). For case 2, wind speeds at a height of 10 m agl were less than 5 m/s over the modelling domain. The mean wind direction was towards the south and south-west near the ground, transitioning towards the south-east up to 1000 m agl, and towards east and north-east above 1000 m agl. The south-north wind component $V$ ranged from -1 m/s to 1.5 m/s in the vertical, with the west-east wind component $U$ ranging from -0.5 m/s near the ground surface to 7.5 m/s up to 2500 m agl. See

Appendix Figure C2 for meteorological vertical profiles for this case. The weak advection and the strong vertical wind shear resulted in tracer plumes being transported for only a few kilometres in the horizontal and mainly staying within the boundaries of the CNRL facility (Fig. S6).

Unstable atmospheric conditions persisted during the simulation period over the region of interest for case 2. Gradient Richardson number ($Ri$) values, which is a measure for atmospheric dynamic stability (Fathi et al., 2021), were below the

critical value of $Ri_c = 0.25$ up to 400 m agl. $Ri$ values below 0.25 correspond to unstable conditions in the atmosphere (AMS, 2022). Consequently, tracer plumes from emission sources mixed in the vertical up to 2000 m during the simulation time. Similar meteorological and atmospheric conditions were observed during the the JOSM 2013 field campaign during the same period. These conditions were also simulated using Environment and Climate Change Canada's (ECCC) air-quality model GEM-MACH at 2.5 km resolution. Our super-resolution (50 m) WRF model simulations were also successful in recreating

the conditions on 26 August 2013 over CNRL. See (section 4.2, Fathi et al., 2021) for detailed discussions on how low and variable wind and unstable conditions affect the atmospheric transport of tracer plumes for this case.

Case 3 model simulations at super-resolutions for the period during 2 September 2013 started at 15UT. Tracer emissions were initiated at 15:30UT. Atmospheric conditions during this case were fairly stable, with Gradient Richardson number $Ri$ exceeding the critical values $Ri_c = 0.25$ (indicating atmospheric dynamic stability) below 100 m above ground level (agl).

Wind speeds were higher for this case compared to the other cases with west-east wind component $U$ ranging from 5 m/s near ground surface to 15 m/s up to 2000 m agl. The south-north component of the wind $V$ was about 5 m/s near the surface, with a strong shear in the vertical transitioning towards south at about 500 m agl. See Figure C3 for meteorological vertical profiles during the time period of this case. Similar to case 1, the mean direction of transport was towards east and north-east, but with a stronger northward component compared to case 1 (see Figure S7).





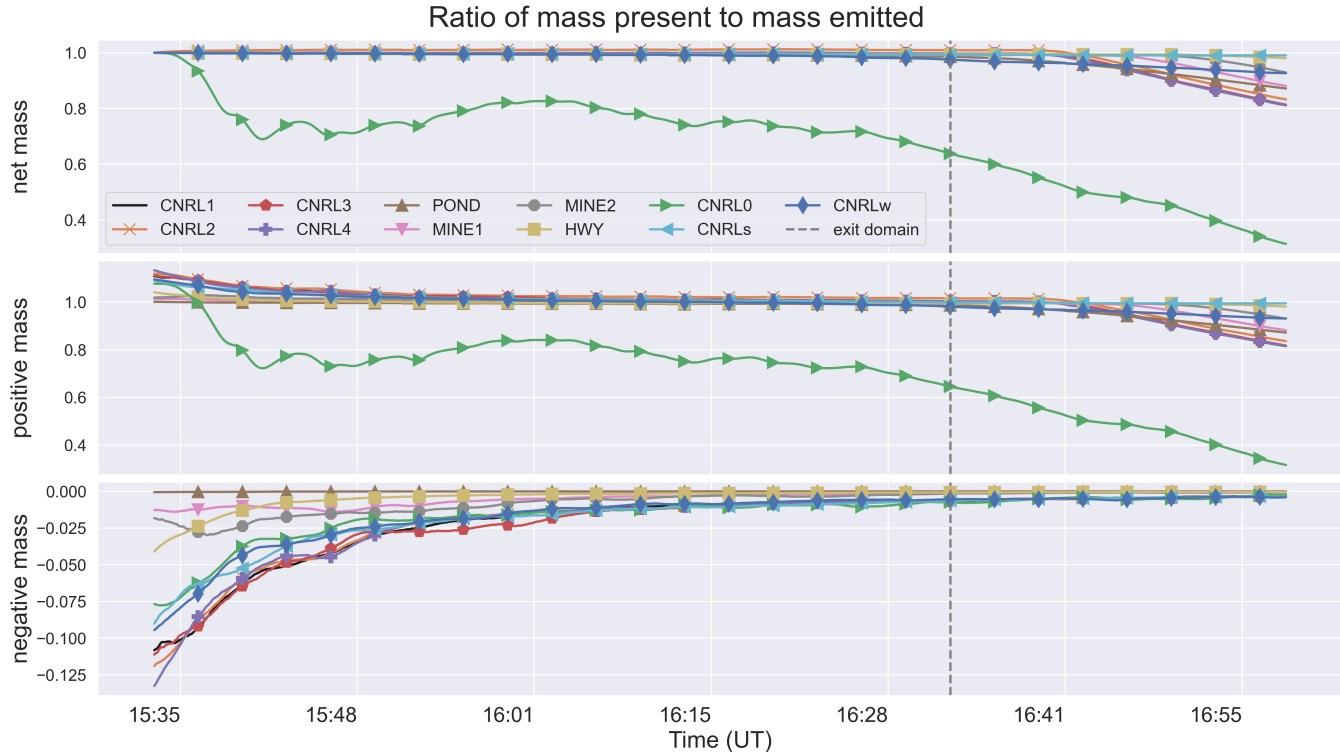

**Figure 8.** The ratio of tracer mass present within the modelling domain to mass emitted over simulation time after the start of tracer release. Time series are separated into positive and negative mass in the bottom two panels. Tracer mass remains conserved over time, with the exception of CNRL0. Note that the drop in present-to-emitted mass ratio beyond 16:30UT, marked by the vertical dashed line, is due to tracer mass exiting through domain boundaries and not a violation of mass conservation.

### 3.3 Model Global Tracer Mass Conservation and Emission Rate Calculation

In this section we evaluate the conservation of tracer mass within domain d05. The Runge-Kutta transport scheme in our WRF modelling setup is conservative, however it does not guarantee positive definiteness on its own. Negative mass creation is offset by positive mass such that tracer mass is conserved over the modelling domain (Skamarock et al., 2008). Negative mass can be set to zero, but this will result in erroneous increase of tracer mass within the modelling domain. By choosing a positive-definite scalar advection option in WRF, as we did in our simulations, a flux re-normalization is applied to the transport step to remove the nonphysical effects such as creation of the negative mass and the excess positive mass (Skamarock and Weisman, 2009). The turbulent diffusion step in the model is also prone to creating negative mass, but to a lesser degree. As we show in the following, the positive-definite re-normalization applied to the advective transport is not always able to filter out the negative mass (and the excess positive mass) created by the diffusion transport. This can result in local mass deficit/surplus.

We investigated the tracer mass budget within domain d05 by integrating over the entire domain at each model output time-step (Eq. 2). Figure 8 shows the ratios of mass present in the modelling domain at each timestamp to the total mass emitted up





to that point. The ratios are for the eleven emission scenarios for the first hour after tracer release was initiated. The net mass present within the modelling domain is separated into negative and positive mass. Time-series are shown as the ratio of mass present in the domain to mass emitted (normalized) for each emission source. Initially up to 14% negative mass (Fig. 8, bottom

panel) is created within the domain which is offset by about the same amount of excess positive mass creation (Fig. 8, middle panel). The net mass (Fig. 8, top panel) is conserved as the negative and the excess positive mass cancel out. The creation of negative mass is reduced over simulation time until it falls below 2% after about 30 min simulation time. The creation of negative mass is likely due to sharp gradients in tracer mass immediately after the initial release, which are concentrated mostly upwind of the emission sources. As tracer mass is advected and dispersed over several grid points, the gradient is smoothed out

and negative mass creation becomes less pronounced. The present-to-emitted mass ratio for 10 out of 11 sources are conserved and remain equal to unity up to about 16:30UT when they reach the domain boundaries and are removed from the modelling domain (Fig. 8, top panel). The exception is CNRL0, which does not keep up with the emissions and, except for a few minutes after the initial release, is always less than one. The mass present in the domain (not normalized) increases initially and later plateaus (approaching an asymptote) as plumes start exiting through domain boundaries at rates less than or equal to source

emission rates. Similarly, the decrease in present-to-emitted mass ratio in Fig. 8 (marked by a vertical line) is a result of tracer mass exiting through domain boundaries at rates less than or equal to source emission rates (and not a violation of model mass conservation).

Figure 9 shows temporal rate of change in tracer mass for each emission scenario, normalized to model input emission rate (MIE). This rate of change was calculated by differentiating the domain mass time-series shown in Fig. 8 according to Eq.

3, which is equivalent to the storage rate term for the entire domain ($S_{C,d05}$). The net rate of change (top panel) is separated into rates for positive and negative mass in the bottom two panels, for each tracer case. The rate of creation of negative mass oscillates between $-10\%$ and $5\%$ and is damped over time. This is offset by the creation of excess positive mass, resulting in net mass conservation over the modelling domain (an artifact of the model transport scheme). For tracer mass to be conserved, the net rate of change over the entire domain must be equal to model input emission rate for the time period before tracer

plumes start exiting through domain boundaries (before about 16:30UT for most tracers as shown in Fig. 9). Tracer mass at domain boundaries was set to zero (boundary condition). As plumes reached the boundaries the tracer mass was removed from the domain (at different times for each tracer depending on the transport speed and source proximity to the boundaries), hence the drop in $S_{C,d05}$ as shown in Fig. 9. This was true for all emission scenarios as can be seen from Fig. 9 top panel. The $S_{C,d05}$ time-series for ten out of eleven sources were at MIE level, indicating mass conservation for these cases. The only exception

was CNRL0 for which $S_{C,d05}$ oscillated around MIE without converging. It is evident from Figures 8 and 9 that CNRL0 mass was not conserved.

For the case of CNRL0, tracer release height was at the model vertical level where model resolution (vertical) transitions from model layer thickness of about 12 m to progressively increasing thicknesses (and decreasing resolutions). Transport (advection and diffusion) of tracer amounts between model vertical layers of varying resolutions, resulted in erroneous and

unbalanced (by excess positive mass) creation of negative mass (both in magnitude and spatial location). The positive-definite transport scheme in our WRF setup failed to conserve the tracer mass for this case, which resulted in about 31% loss. Table 5





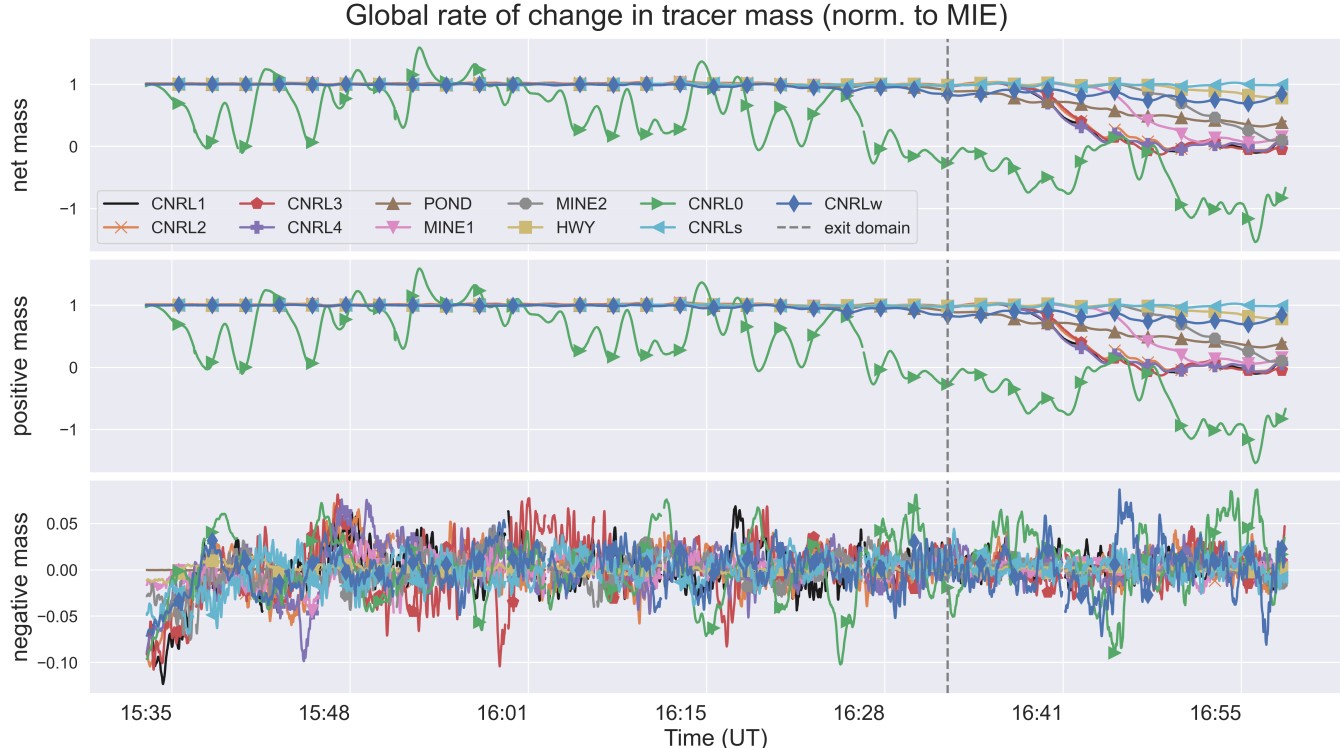

**Figure 9.** Temporal rate of change in tracer mass for the entire domain $S_{C,d05}$ is shown. Time series are normalized to model input emission (MIE) rates. The net rate of change (top panel) is separated into rates for positive and negative mass in the bottom two panels. The creation of negative mass is offset by creation of excess positive mass. Rate of change in tracer mass match MIE (conserved) for all except one emission source (CNRL0).

lists statistics for the first hour of tracer emissions (before exiting through domain boundaries) for all emission scenarios. The temporal means ($\mu$) and standard deviations ($\sigma$) are indicated for the normalized rates ($S_{C,d05}$) shown in Fig. 9. As mentioned before, CNRL0 is the only non-conservative case with a mean value of 0.69 and standard deviation of 0.39. Therefore, in this

paper and in our companion paper (Fathi and Gordon, 2022) we discuss CNRL0 mainly in terms of plume behaviour rather than a conserved quantity in mass-balance calculations. The mean normalized rates for CNRL2 and CNRLw were within 2% and 1% of MIE, respectively. For the remaining 8 out 11 cases, the mean normalized rates were equal to 1.00 (strong agreement with model input emission rates). These results apply to all three case studies within 2-4%.

### 3.4 Local Mass Conservation and Mass-balance Analysis

Results in Table 5 indicate global (over the entire modelling domain) mass conservation for 10 out 11 emission cases. We further evaluated model local mass conservation through mass-balance and flux calculations in order to assess the model's ability to be used for accurate mass-balance assessment. A control volume enclosing all the emission sources, with the downwind wall





**Table 5.** Normalized (to MIE) rates of change in tracer mass for all emission scenarios are shown. Mean rates are given for net, positive (+) and negative (−) mass. Standard deviations are also provided. Rates agree with model input emissions within 2% for 10 out 11 tracers (conserved).

| Tracer ID | Temporal Mean ($\mu$) | | | Standard Deviation ($\sigma$) | | |
| --- | --- | --- | --- | --- | --- | --- |
| | net | + | − | net | + | − |
| CNRL1 | 1.00 | 1.00 | 0.00 | 0.07 | 0.07 | 0.03 |
| CNRL2 | 1.02 | 1.00 | 0.00 | 0.07 | 0.07 | 0.03 |
| CNRL3 | 1.00 | 1.00 | 0.00 | 0.07 | 0.07 | 0.04 |
| CNRL4 | 1.00 | 1.00 | 0.00 | 0.07 | 0.07 | 0.03 |
| POND | 1.00 | 1.00 | 0.00 | 0.07 | 0.07 | 0.00 |
| MINE1 | 1.00 | 1.00 | 0.00 | 0.07 | 0.07 | 0.02 |
| MINE2 | 1.00 | 1.00 | 0.00 | 0.07 | 0.07 | 0.03 |
| HWY | 1.00 | 1.00 | 0.00 | 0.07 | 0.07 | 0.01 |
| CNRLs | 1.00 | 1.00 | 0.00 | 0.07 | 0.07 | 0.03 |
| CNRLw | 0.99 | 0.99 | 0.00 | 0.07 | 0.07 | 0.03 |
| CNRL0 | 0.69 | 0.69 | 0.00 | 0.39 | 0.39 | 0.03 |

at a 5 km distance from the main CNRL stacks, was considered (large rectangle in Fig. 6). We conducted 4D mass-balance calculations using this control volume for all case studies, to evaluate the performance of model simulations in the context of
local mass conservation and transport of tracer amounts within a sub-domain (control volume). The mass-balance calculations for this portion of our analysis were conducted for the period between 1 to 2 hours after the tracer release started, well after plumes crossed the box (control volume) walls.

Mass-balance calculations were done according to Eq. 6. The net flux out term $F_{C,out}$ was calculated using the instantaneous fields (e.g., wind speed, tracer concentrations) along top and lateral boundaries of the control volume. $F_{C,out}$ includes horizon-
tal ($F_{C,H}$, $F_{C,HT}$) and vertical ($F_{C,V}$, $F_{C,VT}$) advective and turbulent fluxes across box walls. The rate of mass storage/release $S_C$ within the control volume (box) at each model output time-step (1 second) was calculated by integrating over the entire volume of the box and differentiating with respect to time. Source emission rates $E_C$ were calculated as the sum of $F_{C,out}$ and $S_C$ and were compared to model input emission rates (MIE). Contributions (absolute value) of different terms are shown in Figure 10 for case 1. The main contribution came from the horizontal advective flux, along with significant contributions from
the storage rate term $S_C$. The horizontal turbulent flux contributed between −0.3% to 2.3% for the three cases. Contributions from vertical fluxes were negligible as the box top was chosen at a height of over 4 km agl, well above the mixing layer height at about 2-3 km agl. The breakdown of contributing terms for the three cases are summarized in Tables 6.

For case 1 (Table 6), mass-balance estimates were within 4% of MIE for 10 out of 11 tracers. Estimates for the (globally) non-conservative tracer CNRL 0 were biased low by 51%. Estimated emission rates for all the other emission scenarios were
in over 96% agreement with model input emission rates (MIE). Contributions from the storage rate term $S_C$ were negative



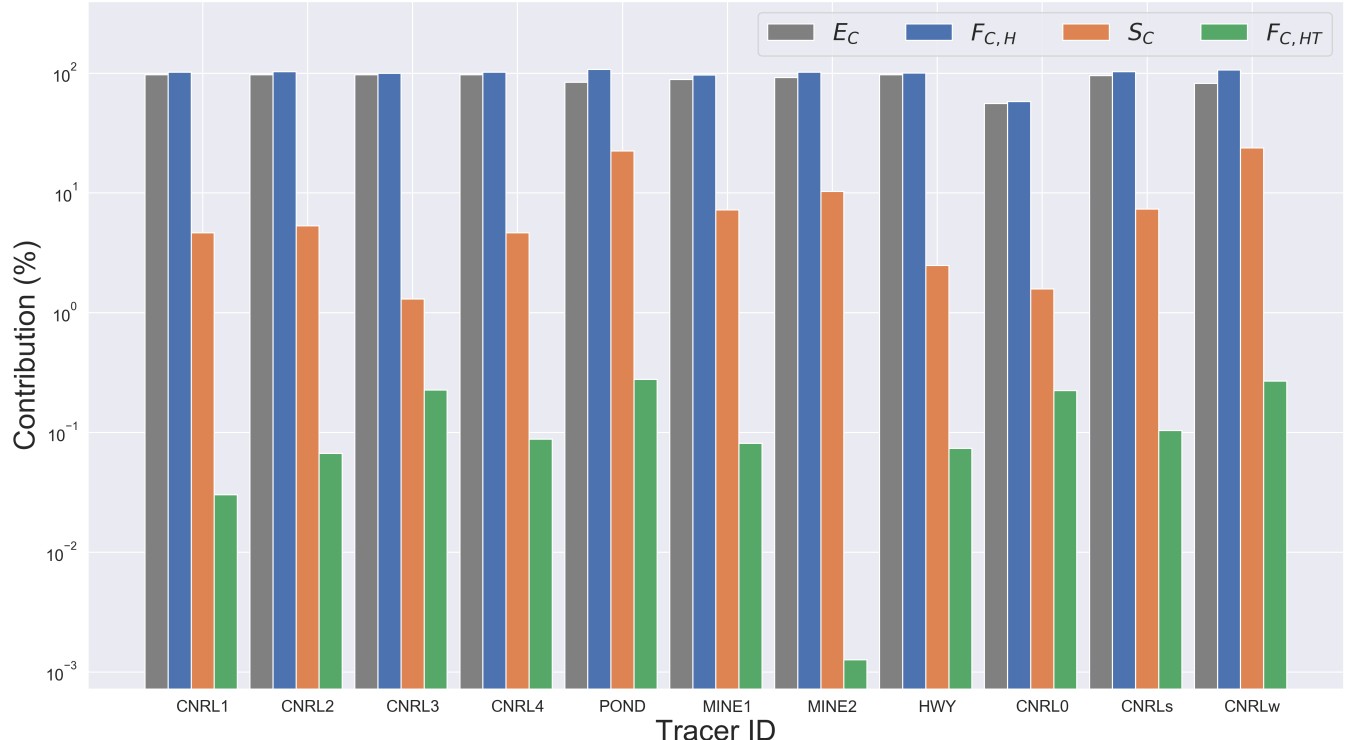

**Figure 10.** Normalized (to model input emissions - MIE) retrieved emission rates for tracer emission scenarios of case 1. Emission rates ($E_C$) were calculated using the mass-balance equation Eq. 6 and the control volume shown in Fig. 6. The breakdown (%) of contributing terms are shown as absolute values, see Table 6 for the actual values. The vertical axis is in log scale. Refer to Table 2 for source specifications.

(release of mass from the control volume) for ten emission scenarios and ranged between 1% to 30%. The contribution of $S_C$ for MINE2 was positive (storage of mass in the control volume) and about 4%. $F_{C,H}$ contributions were positive and ranged between 51% to 130%, which were offset by negative $S_C$ contributions. Our results with GEM-MACH model simulations in Fathi et al. (2021) for $SO_2$ emissions (stack/point source) for the same time period, were very similar (97% $F_{C,H}$ and 3% $S_C$)
to stack emissions simulated here with WRF at super-resolution. The model based study by Panitz et al. (2002) attributes 85% to 95% of the emissions to advective fluxes (for $SO_2$ and CO), which is also consistent with our estimates. Our results here with WRF (LES) simulations are within the range of previous model studies.

Mass-balance estimates for case 2 were within 5% of MIE for the ten emission scenarios (Tables 6). Estimates for CNRL0 were biased low, similar to case 1, by about 50% (not conserved). For this case, where wind speeds were very low and spatially
heterogeneous, the tracer amounts were mainly stalled within the control volume over the simulation time. Consequently, the storage rate term $S_C$ was the dominant term with contributions (positive) ranging from 89% to 103%. The horizontal advective flux $F_{C,H}$ contributed to the mass-balance equation between 0.6% to 12%. The contributions from vertical and turbulent fluxes were less than 1%, except for CNRLw with 2.3% contribution. Our estimates for $SO_2$ emission rates for





**Table 6.** Source emission rates ($E_C$) determined by performing mass-balance calculations on model output data, evaluated against model input emissions (MIE) for the three cases. Results are shown as normalized mean bias (nmb) in %. Contribution of three main terms in the mass-balance equation are shown, normalized to MIE (%). Values less than 0.1% are indicated with $\epsilon$.

| | Case 1 | | | | Case 2 | | | | Case 3 | | | |
| | $E_C$ | Contributions (%) | | | $E_C$ | Contributions (%) | | | $E_C$ | Contributions (%) | | |
| Tracer ID | nmb (%) | $S_C$ | $F_{C,H}$ | $F_{C,HT}$ | nmb (%) | $S_C$ | $F_{C,H}$ | $F_{C,HT}$ | nmb (%) | $S_C$ | $F_{C,H}$ | $F_{C,HT}$ |
|---|---|---|---|---|---|---|---|---|---|---|---|---|
| CNRL1 | -4.0 | -2.9 | 99.2 | -0.3 | -1.7 | 97.7 | 0.7 | $\epsilon$ | 1.5 | -27.5 | 127.3 | 1.7 |
| CNRL2 | -3.2 | -4.4 | 101.2 | -0.1 | 3.3 | 102.7 | 0.6 | $\epsilon$ | 1.3 | -26.6 | 126.3 | 1.5 |
| CNRL3 | -3.7 | -3.7 | 100.0 | -0.1 | -2.6 | 94.6 | 2.8 | $\epsilon$ | 1.0 | -27.5 | 127.1 | 1.4 |
| CNRL4 | -3.8 | -3.9 | 100.2 | -0.1 | -3.9 | 92.4 | 3.6 | $\epsilon$ | 1.4 | -27.2 | 127.0 | 1.6 |
| POND | -4.4 | -10.9 | 106.2 | 0.3 | -4.2 | 92.0 | 3.8 | $\epsilon$ | -3.8 | 15.4 | 80.6 | 0.2 |
| MINE1 | -0.4 | -8.6 | 108.4 | -0.2 | -1.9 | 96.1 | 2.0 | $\epsilon$ | -1.2 | 5.1 | 93.3 | 0.3 |
| MINE2 | 3.9 | -3.0 | 107.3 | -0.3 | 1.6 | 97.4 | 4.2 | $\epsilon$ | -0.9 | 5.6 | 92.7 | 0.7 |
| HWY | -1.2 | -3.0 | 101.8 | $\epsilon$ | -4.8 | 90.3 | 4.7 | 0.2 | -0.6 | -19.2 | 118.3 | 0.3 |
| CNRLs | -3.8 | -8.7 | 105.0 | $\epsilon$ | -4.9 | 90.7 | 4.3 | $\epsilon$ | -1.4 | 10.0 | 88.0 | 0.6 |
| CNRLw | -3.8 | -29.8 | 126.2 | -0.1 | 3.1 | 88.9 | 11.9 | 2.3 | -3.1 | 72.1 | 24.6 | 0.1 |
| CNRL0 | -50.7 | -1.3 | 50.9 | -0.3 | -49.5 | 49.7 | 0.8 | $\epsilon$ | -54.1 | 5.7 | 40.6 | -0.4 |

CNRL with GEM-MACH simulations in Fathi et al. (2021) showed similar large storage levels. We note that conditions of
weak advection were also observed during JOSM 2013 airborne campaign for the period of case 2 on 26 August 2013 (see
Fig. 5), resulting in negligible estimated emission rates based on mass-balance calculations. As a result of such conditions, the
emission estimation flights for this time period were not included (rejected) for top-down retrievals in the relevant published
work. Note that GEM-MACH model simulations at 2.5 km resolution for the same case (26 August 2013), predicted a storage
contribution of only 43% (Fathi et al., 2021). Whereas, WRF super-resolution simulations in this work predicted a contribution
of $\geq 89\%$. While both modelling setups replicated the same meteorological conditions, the coarse resolution (2.5 km) of
GEM-MACH simulations (compared to WRF simulations at 50 m resolution) resulted in larger downwind dispersion of tracer
amounts and larger predicted horizontal mass flux. The WRF super-resolution simulations in this work were successful in
closely replicating the observed weak advection conditions ($F_{C,H} < 20\%$) for the same period, by resolving the transport of
tracer amounts at higher spatio-temporal resolutions. Which demonstrates the benefits of employing super-resolution over high
resolution modelling.

Mass-balance estimates for case 3 were similar to estimates for case 1, with over 96% agreement with model input emissions
(MIE) for the ten emission scenarios. Similar to case 1 and case 2, CNRL0 estimates were biased low (at 54%). For case 3,
the contribution of the storage rate term $S_C$ ranged between -27.5% to 72.1%. The horizontal advective flux $F_{C,H}$ contributed







**Case1, tracer: MINE 1**

**Figure 11.** Three panel snapshot of MINE1 plume during case 1 at 5 min intervals. Note that the erroneous negative mass creation (blue contours) is concentrated upwind of the emission source and on plume edges. Consequently, 98% of the domain total negative mass was located within the control volume along with only 43% of total positive mass (red). The vertical dashed line shows the location of control volume downwind wall.

between 25% to 127% to the mass-balance equation. The horizontal turbulent flux contributed between -0.4% to 1.7%. Similar
to case 1 and case 2, contribution of vertical fluxes were negligible.

As can be seen from Table 6, our mass-balance estimates for the three cases were within 5% of model input emissions (MIE). Estimates were partially affected by local mass deficit/surplus for these cases. As discussed at the beginning of this section,





the turbulent diffusion step in our WRF model setup created erroneous negative mass (locally) within the modelling domain. The negative mass was mainly created near the emission source (upwind) and at plume edges, as shown in Figure 11. The

advection of positive/negative mass passed the box-downwind-wall (vertical dashed line in Fig. 11) was not always balanced, due to different spatial distributions for positive and negative amounts. Fig. 11 shows three snapshots at 5 min intervals for tracer MINE1 during the final hour of case 1. As indicated on the figure, during this time period 98% of the domain total negative mass remained inside the control volume along with only about 43% of the domain total positive mass. This resulted in a mismatch between the negative mass and the excess positive mass and the consequent mass emission rate underestimations

for this and similar cases. Estimates were also partially affected by the changing vertical grid spacing for upper model layers, for tracer amounts mixing to higher altitudes ($> 2.5$ km). This is similar to mass loss for CNRL0 emissions, but to a much lesser extent. We conclude that all three cases (with ten emission scenarios) were globally (on the entire modelling domain) and locally (control volume) mass conserved within 4% and 5% of MIE , respectively.

## 4   Conclusions

We developed and implemented super-resolution ($< 100$ m) model simulations employing the WRF-ARW atmospheric modelling system. We used NARR reanalysis data at 31 km resolution as initial and boundary conditions for our coarsest domain at the same resolution. Dynamical down-scaling of reanalysis data was done through numerical model nesting with five domains from 31 km to 50 m at the finest resolution domain over the oil sands facility CNRL. We chose our model simulation times and locations from three emission rate retrieval flights during the JOSM field campaign in August and September 2013. We

performed model simulations for three days in August and September, representing different meteorological conditions. Model simulations for the finest resolution domain were conducted using Large Eddy Simulation (LES) sub-grid parameterization. The main objective was to model the state of the atmosphere at high enough resolution to simulate atmospheric dynamical processes at spatial and temporal scales of airborne measurements, while ensuring local and global mass conservation. Model output data from our simulation cases were evaluated against historical observational data from two WBEA monitoring sta-

tions, as well as aircraft observations during 2013 JOSM campaign for the same locations and time periods. Model output fields showed good agreement with observational data within 5% in terms of 2-m temperature, 2-m relative humidity, and 10-m winds. General wind directions generated by the super-resolution model were within 20 to 30 degrees of the observational data. Model wind speeds were in agreement with aircraft observations within one standard deviation.

We modified WRF model dynamical solver source code to simulate passive tracer emissions within the finest modelling

domain, centred over the CNRL facility. Our case simulations included eleven different emission scenarios with stack/point and surface/area sources. Continuous emissions were simulated for time periods of 2-3 hours during three different days. We evaluated the atmospheric dispersion and transport of tracer plumes from the eleven emission scenarios under different meteorological conditions. Unstable atmospheric conditions, low wind speeds, and strong vertical wind shear during our case study on 26 August 2013 resulted in weak advection of tracer plumes during the simulation time. During case studies on 20

August and 2 September, atmospheric conditions were relatively more stable with higher wind speeds of about 5-15 m/s. Tracer





plumes from emission sources were advected mainly towards east and north-east for these two cases. Similar conditions have been observed during the JOSM 2013 field campaign for the same time periods and locations as our three case studies.

We evaluated the performance of our model simulations in terms of global (over the entire domain) mass conservation. For one case out of eleven, creation of erroneous mass by the model transport step resulted in loss of tracer mass of about 30%.

For this emission case, tracer release was placed at the model vertical level where model vertical resolution transitions from the super-resolution of about 10 m to progressively coarser resolutions. The unbalanced creation of erroneous mass at sharp concentration gradients, such as at the vicinity of point sources, is intensified on an irregular grid (an artifact of model transport scheme). The positive-definite re-normalization scheme in our modelling setup was successful in preventing such nonphysical effects for 10 out 11 emission sources (assigned on a regular grid). The rate of change for tracer mass was calculated by

integrating over the entire modelling domain and differentiating with respect to time. Results were within 2-4% of model input emissions (MIE) for 10 out of 11 emission scenarios, indicating global mass conservation in our simulation cases. Therefore, it is recommended that tracer emissions be assigned to model grid points with regular grid spacing for several adjacent model cells along the $x$, $y$, $z$ directions.

We further investigated local mass conservation by mass-balance calculations over a sub-domain (control volume). Mass-

balance calculations were conducted by considering a control volume (box) enclosing all emission sources. We estimated within-box source emission rates by calculating the net exiting flux through box top and lateral walls, and the temporal rate of change in tracer mass within the control volume. Under normal advective conditions ($> 5$ m/s wind speeds), the horizontal advective flux was equal to over 90% of the emission rate while turbulent fluxes were less than 2%. The remaining contribution came from the storage rate term, the release/accumulation rate of tracer mass within the control volume. Our mass-balance

estimates were within 5% of MIE for tracer sources with release heights within the bottom 40 model vertical layers (with regular vertical grid spacing) for our three case studies (30 emission scenarios), indicating local mass conservation for our super-resolution WRF simulations. Again, this suggests that assigning tracer release on a regular grid (horizontal and vertical) would ensure mass conservation using the available WRF model mass conservation schemes.

Except for one hypothetical emission source, our results for various tracer emission scenarios under different meteorological

conditions were globally and locally mass conserved within 4% and 5% of model input emission rates, respectively. In our companion paper (Fathi and Gordon, 2022) we use the model output from super-resolution WRF simulations discussed here for a model-based study of airborne top-down source emission rate retrievals. We evaluate the conventional methods and propose improved approaches for aircraft-based retrievals.

*Code and data availability.* The release version of WRF-ARW 3.9 used for this study can be downloaded from https://www2.mmm.ucar.edu

/wrf/users/download/get_source.html. The North American Regional Reanalysis (NARR) data from National Oceanic and Atmospheric Administration (NOAA) used as initial and boundary condition for model simulations in this study can be accessed at NOAA-Fathi (2022). The historical observational monitoring data from Wood Buffalo Environmental Association (WBEA) used for comparison to model output





in this work can be accessed at WBEA-Fathi (2022). The aircraft measurements data from JOSM 2013 campagin used in this work are avaibale from Environment and Climate Change Canada Data Catalogue (ECCC, 2013).

**Appendix A: Prognostic TKE Closure**

The WRF model eddy viscosity (diffusivity) for the predicted turbulent kinetic energy option (K option *km_opt*= 2), are computed using (Skamarock et al., 2008),

$$K_{h,v} = C_k l_{h,v} \sqrt{e} \tag{A1}$$

where $e$ is the turbulent kinetic energy (prognostic in this scheme), $C_k$ is a constant (typically $0.15 < C_k < 0.25$), and $l$ is a
length scale given as follows for the anisotropic option,

$$l_h = \sqrt{\Delta x \Delta y} \tag{A2}$$

and,

$$l_v = \quad \min \left[ \Delta z, 0.76 \sqrt{e}/N \right] \quad \text{for } N^2 > 0 \tag{A3}$$
$$l_v = \quad\quad \Delta z \quad\quad\quad\quad \text{for } N^2 \leq 0. \tag{A4}$$

where $N$ is Brunt-Väisälä frequency, see Skamarock et al. (2008) for derivations. The eddy viscosity used for mixing scalars is divided by a turbulent Prandtl number $P_r$ . The Prandtl number is $1/3$ for the horizontal eddy viscosity $K_h$ , and $P_r^{-1} = 1 + 2l/\Delta z$ for the vertical eddy viscosity $K_v$. Note that the above are for the anisotropic mixing option (mix_isotropic = 0, default) that was used in our WRF simulations.

**Appendix B: Mass-balance turbulent flux terms**

Derivations for turbulent flux terms in mass-balance equation Eq. 1,

$$F_{C,HT}(t) = - \iint K_h \frac{d\chi_C}{dx_\perp}(t,s,z) ds dz \tag{B1}$$

$$F_{C,VT}(t) = - \iint K_v \frac{d\chi_C}{dz_\perp}(t,x,y) dx dy \tag{B2}$$

where $K_h$ and $K_v$ are horizontal and vertical eddy diffusivity coefficients, respectively (see A). $d\chi_C/dx_\perp$ and $d\chi_C/dz_\perp$ are tracer concentration gradients across box lateral and top walls.





**Appendix C: WRF model output meteorology profiles for the three case studies**

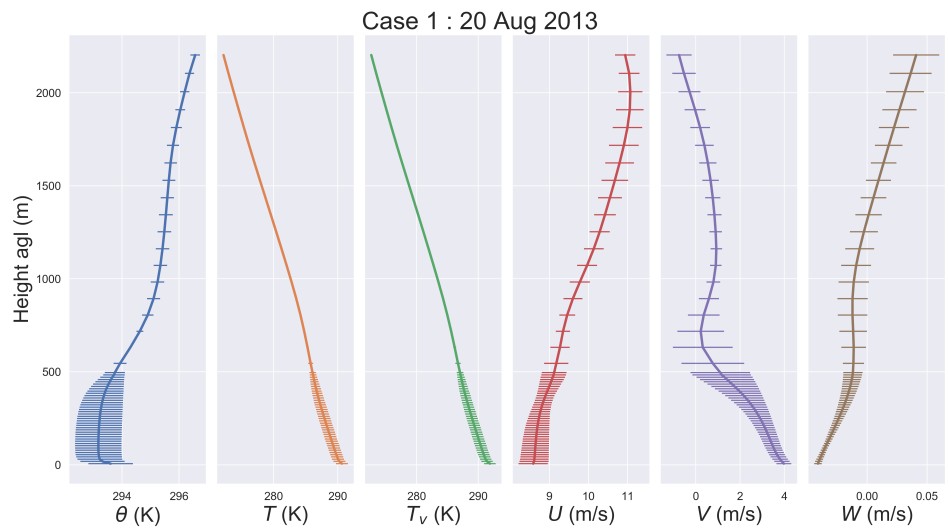

**Figure C1.** Case 1 mean meteorological profiles on 20 August 2013: potential temperature $\theta$, absolute temperature $T$, virtual temperature $T_v$, west-east wind $U$, south-north wind $V$, and vertical wind $W$. Bars show standard deviations during the simulation time.

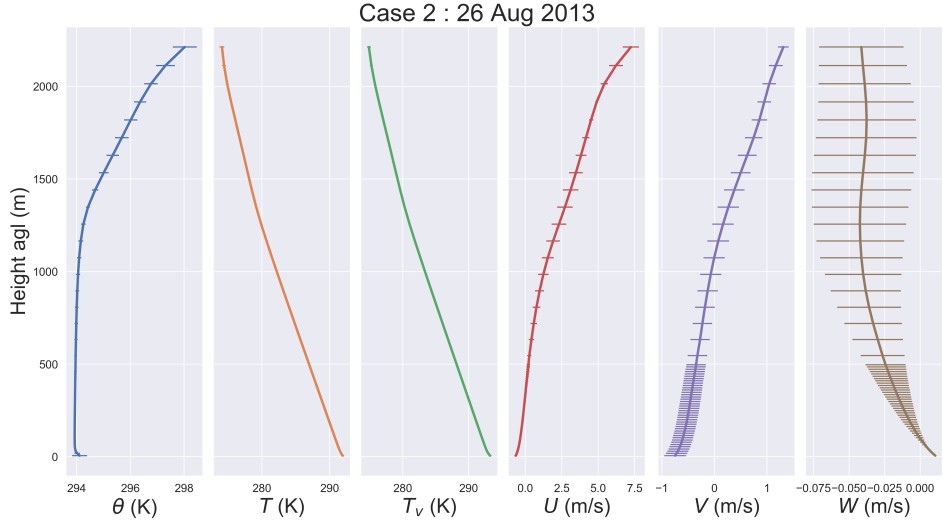

**Figure C2.** Case 2 mean meteorological profiles on 26 August 2013: potential temperature $\theta$, absolute temperature $T$, virtual temperature $T_v$, west-east wind $U$, south-north wind $V$, and vertical wind $W$. Bars show standard deviations during the simulation time.





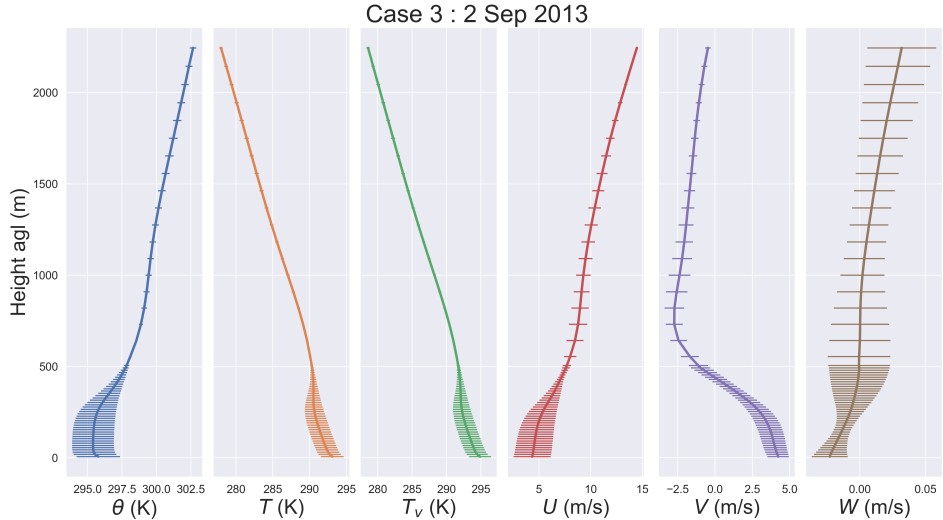

**Figure C3.** Case 3 mean meteorological profiles on 2 September 2013: potential temperature $\theta$, absolute temperature $T$, virtual temperature $T_v$, west-east wind $U$, south-north wind $V$, and vertical wind $W$. Bars show standard deviations during the simulation time.



*Author contributions.* SF setup and ran the WRF-ARW simulations, performed all the analysis using the model output data, and wrote the paper. MG and YC provided advice during planning and analysis and contributed to paper revisions.

*Competing interests.* The authors declare that they have no conflict of interest.

*Acknowledgements.* This research was enabled in part by support provided by Compute Ontario (Graham, graham.computecanada.ca),
WestGrid (Cedar, cedar.computecanada.ca) and the Digital Research Alliance of Canada (alliancecan.ca). We acknowledge use of flight parameter information from the ECCC Data Catalogue: Pollutant Transformation, Summer 2013 Aircraft Intensive Multi Parameters, Oil Sands Region.





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
