# Peer review of "Passive Tracer Modelling at Super-Resolution with WRF-ARW to Assess Mass-Balance Schemes"

_EGUsphere, 2022_

## Author Comment (AC1)

**EGUSPHERE-2022-1125 | Development and technical paper**

Submitted on 20 Oct 2022

**Passive Tracer Modelling at Super-Resolution with WRF-ARW to Assess Mass-Balance Schemes**

Sepehr Fathi, Mark Gordon, and Yongsheng Chen

**Handling topical editor**: Jinkyu Hong, jhong@yonsei.ac.kr

**Author Responses to Referee Comments**

**Color Code**
*RC, Referee Comments in Black*
*AC, Author Comments in Blue*

**Comment on egusphere-2022-1125**

Anonymous Referee #1

Referee comment on "Passive Tracer Modelling at Super-Resolution with WRF-ARW to Assess Mass-Balance Schemes" by Sepehr Fathi et al., EGUsphere, https://doi.org/10.5194/egusphere-2022-1125-RC1, 2023

RC1_01: This work features high-resolution numerical simulations with the model WRF-ARW using realistic meteorological boundary conditions and employing a multistep nesting approach. A tracer transport module is used to simulate the dispersion of various emission sources at the finest LES scale. Ground-based and airborne observations are used to evaluate meteorological parameters, which generally showed satisfactory performance, though some questions regarding the wind-speed evaluation remain. Finally, the mass-balance scheme was applied on 4-D model output to infer source emission rates and to evaluate local and global mass conservation of the tracer transport scheme. This part of the study can still be better elaborated as suggested by the comments given below. The structure of the manuscript, the quality of the figures, and the presentation of the data and results can be further improved.

AC1_01: We thank you for your valid and thorough comments. We made several revisions in response to your comments which helped improve our manuscript. Specific responses to each comment are provided below.

**Major comments**

RC1_02: The authors evaluate the performance of the WRF tracer transport scheme regarding mass conservation and the positivity of transported concentrations. I think since this is one of the most relevant results of this study, a more detailed description of the transport schemes in the paper is necessary. What is the differential equation to be solved? How are the advection and diffusion terms discretized in space and time? What kind of flux redistribution is applied for positivity? The authors state that the turbulent diffusion step is mainly responsible for the observed negative mass creation. This seems not very plausible unless it is shown by example. Did the authors test the transport scheme with diffusion turned off? The transport scheme is supposed to be conservative (based on the flux- divergence formulation). How is it then possible that a vertically changing grid spacing results in a violation of mass conservation? This needs further clarification.

AC1_02: Thank you for raising these valid and important points. Additional discussions were added throughout (especially to Section 2) to expand on descriptions for transport schemes, differential equations, advection and diffusion, positive definiteness, negative mass creation and mass conservation. Regarding the diffusion scheme, we included the following in the revised manuscript section 2.2: "Various formulations are available in ARW solver for explicit spatial diffusion (turbulent mixing) including a sixth order spatial filter proposed by Xue (2000). The implementation of this scheme in ARW is described in Knievel et al. (2007). The sixth order turbulent diffusion scheme is also prone to creating negative mass due to negative up-gradient diffusion. Monotonicity can be enforced in the model (user specified option) by setting negative diffusive fluxes to zero, however it does not conserve scalar mass (Shamrock et al., 2008). Hence, in our simulations we used the sixth order diffusion scheme without the monotonic option." Also, the new Figure S10 was added to show the comparison of the monotonic and non-monotonic diffusion options in WRF. We chose the non-monotonic option due to the fact that the monotonic diffusion, while positive, is not necessarily conservative. The following text was added to first paragraph of Section 3.4 "As discussed in Section 2.2, in our simulations we used a positive-definite transport scheme combined with a sixth order diffusion scheme. Negative up-gradient diffusion flux near sharp concentration gradients resulted in partial creation of erroneous mass within the modelling domain. Positive flux and monotonicity can be enforced in the model by setting negative fluxes to zero, however this is not mass conserved (see Figure S10). Therefore, we configured our simulation using the diffusion scheme without the monotonic option.

[Figure]

Figure S10. Comparison of WRF-ARW default non-monotonic diffusion scheme (left column) with the monotonic-positive diffusion scheme (right column), as the ratio of mass present in the modelling domain to mass emitted.

RC1_03: The implementation details of the mass-balance technique are still only poorly described. The authors should give some details on how the advective and diffusive fluxes ($F_{C,H}$, $F_{C,H,T}$, $F_{C,V}$, $F_{C,V,T}$) at the flux-box boundaries are computed in discrete form. How are the scalar values, which are defined at mass points, interpolated/reconstructed at the cell faces, where the velocity components are defined? Does the offline computation of the fluxes based on model output match the flux computation of the WRF advection/diffusion scheme?

AC1_03: Table S1 was added to the supplement for discrete integral expressions of terms used in the mass-balance equation. The following text was included in Table S1 caption "Model wind fields were linearly interpolated onto mass grid-points for the discrete integral calculations." Further, the following text was added to the last paragraph of Section 2 "See Table S1 for discrete integral expressions of different terms in the mass-balance equation (Eq.6). Note that for flux calculations in this work, model wind fields were linearly interpolated onto the mass grid-points (where concentration fields are defined)."

**Table S1.** Discrete integral expressions of Eqs. (3), (4), (5), (B1) and (B2) to be substituted in Eq. (6). Constants $\Delta r$, $\Delta y$ and $\Delta s$ are equal to model grid resolution in the horizontal, 50 m. $t$ is the time index; $i$, $j$ and $k$ are the 3D-space indices and $s$ is the path index around the control volume (box). Model wind fields were linearly interpolated onto mass grid-points for the discrete integral calculations. Throughout, the second-order central finite-difference scheme was used to numerically solve the time derivatives ($\Delta/\Delta t$) with the residual error of order $O(\Delta t^2)$, where $\Delta t$ is equal to the 1 sec model output time-step.

| Term | | Discrete Numerical Integration |
|------|---|--------------------------------|
| $S_C^t$ | $=$ | $\Delta r \Delta y \sum_{i,j,k}^{n_x, n_y, n_z} \frac{\Delta \chi_{C,ijk}^t}{\Delta t} \Delta z_k$ |
| $F_{C,H}^t$ | $=$ | $\Delta s \sum_{s,k}^{n_s, n_z} \chi_{C,sk}^t U_{\perp,sk}^t \Delta z_k$ |
| $F_{C,HT}^t$ | $=$ | $-K_\Lambda \Delta s \sum_{s,k}^{n_s, n_z} \frac{\Delta \chi_{C,sk}^t}{\Delta x_\perp} \Delta z_k$ |
| $F_{C,V}^t$ | $=$ | $\Delta r \Delta y \sum_{i,j}^{n_x, n_y} \chi_{C,ij}^{t,top} W_{ij}^{t,top}$ |
| $F_{C,VT}^t$ | $=$ | $-K_\tau \Delta r \Delta y \sum_{i,j}^{n_x, n_y} \frac{\Delta \chi_{C,ijk}^t}{\Delta z_\perp}$ |

RC1_04: The horizontal grid-refinement ratio of 1:5 in the nesting process is quite large (typically 1:2 or 1:3 is used). Are there any specific reasons for using such a large grid-refinement ratio? Are there any tests available from which nesting errors could be estimated? How large are the vertical grid refinement ratios near the ground? âz for d03 and d04 are missing in the paper.

AC1_04: The WRF user guide recommends horizontal nesting ratios of up to 1:5, which is what we used as a compromise between computational cost of our simulations and achieving the desired resolution of 50m in the finest domain from the 31km resolution input reanalysis fields. The following text was added to section 2.2: "With regards to nesting in the horizontal, Shamrock et al. (2008) recommended the use of odd nesting ratios such as 1:3 and 1:5 (as opposed to even ratios like 1:2), due to the staggered structure of the Arakawa C-grid used in WRF-ARW modelling framework. Mohan and Sati (2016) investigated the impact of different nesting ratios in WRF and found no statistically significant difference in simulated results with ratios 1:3, 1:5, and 1:7, suggesting that larger ratios can be used to reduce the computational cost in nested simulations. However, larger ratios (e.g., 1:9) can result in increased interpolation errors and are not recommended. In this work, as a compromise between numerical accuracy and computational cost, we used a 1:5 nesting ratio". Also, vertical grid refinement ratios were added to Table 3 "d04: d03 grid refined (~1:3) below 2500 m agl, d05: d04 grid refined (~1:7) below 540 m agl".

RC1_05: Regarding the wind speed evaluation with WBEA observations: The sensitivity of wind speed to model resolution was not satisfactorily explained. It is not convincing that the nesting error would result in a systematic increase

in wind speed at each nesting step. Such a conclusion certainly cannot be drawn from Daniels et al. (2016). This brings me to the question of whether the vertical interpolation of model data to the measurement height of 10m was based on a proper logarithmic law? A comparison of vertical profiles of wind speed between domains d03, d04, and d05 would be helpful to further investigate this large apparent sensitivity.

For model output from domain d05, the first two model layers were at 6m agl and 18m agl, and so the 10m agl wind was determined by interpolating between these to levels. Model output data from domains d03 and d04 included diagnostic variables for U and V at 10m agl. See the newly added Figure S2 where wind profiles from domains d03, d04, d05 are compared. Further, points 2 and 3 in section 3 (old section 3.1 / new section 3.2) were revised as follows:

"2. The wind speeds in the NARR reanalysis data (at 31 km resolution) used as input for our simulations, were higher than WBEA observed values by 2-3 m/s for the region and the periods of interest. Consequently, the bias in NARR winds was carried through model nested simulations.  If replicating the observed atmosphere is an objective of the modelling, it is recommended that input data (e.g., NARR reanalysis) to be adjusted to observations first.

3. Dynamical down-scaling of NARR reanalysis data from 31.25 km resolution to 50 m resolution with five nested domains and vertical grid refining is another source of uncertainty. In concurrent grid nesting as used in this work, output from parent domain is interpolated to provide initial and boundary conditions for each respective nested domain. Horizontal, vertical and temporal interpolation errors are therefore compounded with each nesting. This can result in biased wind fields as in Daniels et al. (2016), which is consistent with our results where d04 wind speeds were higher than d03 by about 1 m/s, and d05 winds were higher than d04 by about 1-2 m/s (see Table 4 for d04 vs. d05). While there may be a relationship between nesting a wind speed error, these results do not directly demonstrate a change in wind speed due to nesting."

RC1_06: Figure 5: Did the authors really show the full vertical extent of model data from the first model layer above ground? If so, it is strange that the wind speed in the model does not decrease towards the ground. The WEBA data shows consistently lower wind speeds than the aircraft data, which obviously reflects the logarithmic wind law within the surface layer. Your model results give no hint of that.

AC1_06: Note that the profiles depicted in Fig. 5 are averaged profiles over the horizontal extent of domain d05 and over the simulation time. Bars show the respective standard deviations (spatial + temporal). Further, Fig. 5 was

revised to show data as a function of height above ground level (agl), and to include aircraft data for the box flight portion of the sampling, which reduces (but does not eliminate) the increase at the surface. Figure S2 is added, where wind profiles (not averaged spatially or temporally) from domains d03, d04, d05 are depicted. These profiles show decreased wind speeds towards ground for all three domains. The following text was added to last paragraph of section 3.2: "Model data were averaged horizontally over domain d05 and the simulation time (Fig. 5), for comparison to aircraft data that were collected during 1-2 hours flight time over the oil sands region. Note that the near surface increase in wind speeds in Fig. 5 is the result of averaging over varying (complex) topography (see Figure S2 for instantaneous profiles at the location of the main CNRL stack). "

[Figure]

Figure S2. Wind vertical profiles from domains d03, d04 and d05 compared at the location of the main CNRL stack for case 1 on 20 Aug 2013 at 18 UTC.

RC1_07: Figure 7 alone provides a very weak basis for the discussion of the vertical mixing characteristics of the plumes. Some features the authors describe (like the effect of the Athabasca river basin) may be just attributable to random turbulent fluctuations. I suggest providing at least an additional plot for temporal mean turbulent statistics, like the average vertical turbulent flux of tracer mass or turbulent stresses.

AC1_07: Further discussion on the role of turbulent mixing was added to the end of Section 3.3.

RC1_08: The structure of the manuscript needs to be improved:

Section 2.2: I suggest separating the model description from the technical setup more clearly. The paragraphs do not seem to follow any clear logic. I would suggest the following paragraphs:

A concise description of the WRF model
The tracer module should be described in more detail (governing equation + spatial discretization), as it is the main focus of the study.
Introduction of the simulation domains d0-d5 with the static information (horizontal + vertical resolution, horizontal extent, coverage) along with the nesting technique Simulation settings of domains d0-d5 + driving data of coarsest domain d0 Implementation of the point and area emission sources in d05

AC1_08: Section 2.2 was revised accordingly. The following text was added to the first paragraph "The Advanced Research WRF (ARW) solver features a suite of fully-compressible Euler-nonhydrostatic equations for solving prognostic variables including velocity components in Cartesian coordinates (u, v ,w), and scalars such as water-vapour mixing ratio and tracer concentration. The 3rd order Runge-Kutta scheme (RK3) is used for time integration in ARW (Wicker and Skamarock, 2002). The spatial discretization in WRF-ARW uses a Arakawa C-grid staggering with thermodynamics/scalar variables (e.g., moisture, tracer) defined on grid cell centres (mass points), and velocity components defined normal to respective faces of model grid cells (one-half grid length from mass points). 2nd to 6th order spatial discretization and RK3 time-integration scheme are available in ARW to solve for advection of momentum, scalars, and geopotential in flux form (the governing equations). The RK3 transport/advection (combined with flux divergence) in WRF-ARW is conservative, however it does not guarantee positive definiteness on its own. Negative mass creation is offset by positive mass such that tracer/scalar mass is conserved over the modelling domain (Skamarock et al., 2008). Negative mass can be set to zero, but this will result in erroneous

increase of tracer mass within the modelling domain. By choosing a positive-definite scalar advection option in WRF-ARW, as we did in our simulations, a flux re-normalization is applied to the transport step to remove the nonphysical effects such as creation of the negative mass (Skamarock and Weisman, 2009). To summarize, if the outgoing fluxes (removing mass from the control volume) in the final step of RK3 predict a negative updated scalar/tracer mixing ratio, the outgoing fluxes are re-normalized to be equivalent to mass within the volume. For more details see Section 3.2.3 in (Skamarock et al., 2008). Various formulations are available in ARW solver for explicit spatial diffusion (turbulent mixing) including a sixth order spatial filter proposed by (Xue, 2000). The implementation of this scheme in ARW is described in Knievel et al. (2007). The sixth order turbulent diffusion scheme is also prone to creating negative mass due to negative up-gradient diffusion. Monotonicity can be enforced in the model (user specified option) by setting negative diffusive fluxes to zero, however it does not conserve scalar mass (Skamarock et al., 2008). Hence, in our simulations we used the sixth order diffusion scheme without the monotonic option."

RC1_09: Section 3.1: The authors should first provide a complete and purely descriptive comparison of model data (domain d05) with WEBA and aircraft observations before they draw any further conclusions regarding model performance. The content from line 310 until the end of the section describes in fact only model sensitivity and does not provide a model evaluation at the location of the oil-sand facility. From line 321 onwards the authors just mostly repeat the content of Table 4, which is quite confusing to read. I would suggest dedicating this paragraph to model sensitivity, labeling it as such, and presenting the data in a more structured way.

AC1_09: Revisions were made accordingly with the discussions on model sensitivity gathered in a new "Section 3.1: Model Sensitivity" and, the Section 3.2 dedicated to model evaluation against observational data.

RC1_10: Section 3.3: It is started here again with a model description. The authors should move this first paragraph to Section 2.2, where the tracer module is introduced.

AC1_10: Revised accordingly, relevant text was moved to Section 2.2.

RC1_11: Many figures need revision:

The subplots of multipanel figures should be labeled in alphabetic order and referenced as such in the caption. The caption should only contain precise and concise descriptive information and no evaluative comments. The caption of Fig. S2 is incomplete and contains an evaluative comment. What

exactly is "level tracer count sum" in Figure 7, S4, and S5? The authors should only use terms that are clearly defined.

AC1_11: Fig. S2 (S3 in the revised manuscript) caption was revised to "Model output meteorological fields from the two fine resolution domains d04 and d05 were evaluated against domain d03 output for the location of CNRL facility in terms of root mean square error (rmse) and mean bias (me). Time series are shown for wind speed at 10 m agal (a, b), temperature at 2 m agl (c), relative humidity at 2 m agl (d), and seal level pressure (e)." Fig 7, S4, and S5 captions were revised "Data shown is the tracer amount summed in the horizontal level and normalized to the maximum value for each source." Color labels for these figures were changed to "Total Mass in Horizontal".

RC1_12: The colorbars of Figures 11 and S1 are still missing. The colorbars of Figures 7, S3, S4, S5, S6, and S7 can be improved (logarithmic scaling of ticks and larger tick labels).
It is difficult to see the wind barbs in Figures S3, S6, and S7, and the thin contour lines (I assume this is terrain height?) further interfere and do not provide further useful information.

AC1_12: Text was added to Fig 11 caption "Colour darkness is proportional to log of column total tracer mass." Figures 7, S3, S4, S5, S6, and S7 were revised accordingly. Colorbar was added for Fig 11.

RC1_13: The labeling of the time axis in Figures 4 and S2 needs to be revised (what is the "20" in front of the time stamp?). Please also include bias values in each subplot of Figure S2, like done for rsme.
For a better comparison of data in Figure 3, I suggest using the same percentage axis (e.g., 0, 10, 20, ..., 60) in the wind-rose plots.

AC1_13: Figures 3, 4 and S2 were revised accordingly.

RC1_14: The tick labels of the x- and y-axis need to be increased in Figures C1-C3. What simulation domain is presented in Figures C1-C3, and is the depicted data horizontally averaged?

AC1_14: Data shown in Figures C1-C3 are from domain d05. The depicted data is horizontally averaged over domain d05 and averaged over time. Figures were revised accordingly. The following text was added to figure captions "The profiles were averaged horizontally over domain d05 and over the simulation time. Bars show standard deviations."

**RC1 Minor comments**

Line 29: missing comma before "which"

Revised

Line 51: "were successful"

Revised

Line 65: Please provide also an estimate for the sampling frequency besides the flying speed. How one gets dx ≤ 100m and dt ≤ 1s?

Revised, text added "e.g., onboard instrument ≥ 1Hz:…"

Lines 85-86: "from the Joint Canada-Alberta Implementation Plan on Oil Sands Monitoring airborne campaign (JOSM 2013)."

Revised

Line 114: "are considered".

Revised

Line 123: "large area rectangular surface source" is a bit strange. It should be clear to the reader that it is a surface source. Maybe "large rectangular area source" is better? Please check also other instances in the manuscript, e.g., "multi-section line surface source" (line 124).

Revised to "large rectangular area (surface) source" and "multi-segment line (surface) source"

Table 2: Please provide an extra column for "Size" for the numbers ~20 km, ~50 km$^2$, etc. Done, the "Spatial extent" column was added.

Caption of Figure 1: Is Hwy an abbreviation for highway? Yes, revised to "highway"

Line 142: This is not a complete sentence. Suggested correction: "…simultaneously. The process where the coarse "parent" domain's output is interpolated to provide initial and lateral boundary conditions for the fine "child" domain is referred to as one-way nesting." Revised

Line 149: "and this is" Done

Line 153: Please either consistently use "JOSM 2013" or "2013 JOSM" throughout the manuscript. Done

Caption of Table3: What exactly are "Coarse" and "Fine" for the vertical grid? Please provide a clear explanation with numbers here or in the main text. According to the main text, "Fine" is not exactly the same for d04 and d05? What is $Z_{top}$? Please use precise language and introduce every variable properly.

The column "vertical grid description" was added to Table 3. Table 3 caption's last sentence was revised to "With model top layer at 15.623 km (15.350 km agl) and pressure level of 10 kPa for all domains".

Line 165: "in the figure Figure 2b."

Line 167: "complex flow conditions" Revised

Line 169: Please provide a reference with section numbers for "In the following sections". Done

Lines 170-175: This paragraph deals again with vertical model resolution (already introduced before line 160) and contains repeated information (see line 158).

The repeated information was removed and the rest of the discussion in the paragraph was moved to lines 159-162 "Note that the vertical resolution of about 12 m for the bottom 500 m agl (above ground level) is sufficient for investigating and evaluating different methods for extrapolating sampled data below the lowest flight level…".

Line 183: Is this really part of the subgrid-scale parameterization or just numerical diffusion to remove spurious small-scale noise of the advection scheme? What exactly is the default option?

Revised to "In order to simulate small-scale atmospheric dynamical processes, the finest two model domains (d04 and d05) were configured with the following dispersion scheme, and Large Eddy Simulation (LES) sub-grid parameterization options available in the WRF model…"

Line 185: What exactly are these modifications? I checked Blaylock et al. (2017) and could not find any WRF modifications related to tracer transport.

Text was revised to "To simulate the emission of passive-tracers, the WRF dynamical-solver source code was modified following an approach similar to Blaylock's (as described in Blaylock 2017) used in Blaylock et al. (2017)."

Reference: Blaylock, B. K.: Tracer Plumes in WRF (last accessed: March 8, 2023), https://home.chpc.utah.edu/~u0553130/Brian_Blaylock/tracer.html, 2017.

Line 205: Please provide the section number here ("...are discussed in Section xx"). Done

Line 308: "were more severe is larger" Revised

Line 308: "We discuss later" Where exactly? Revised "(Section 3.4)"

Line 311: "due to the fact that" Revised

Line 311: "(see above)" Where exactly this is shown? Revised to "(see discussions at the beginning of Section 3.1 and also see Figure 3)"

Line 339: "below" Where exactly (Section xx)? Revised "discussed in Section 3.3"

Line 361: What is an "emission flight"? Revised to "box flight (see Section 2.1)

Line 361: "in relevant publications" Which publications? Only Fathi et al. (2021)? Revised to "was rejected for emission rate calculations due to low and variable wind speeds as reported in Fathi et al. (2021)"

Line 391: If I am not wrong, mass conservation is determined by the spatial discretization (flux-divergence formulation) and is not directly related to the Runge-Kutta scheme, which is just a time integration scheme.

The Runge-Kutta scheme is the transport scheme used in our WRF simulations which ensures conserved transport of tracer amounts the model transport step as described in Shamrock et al., 2008, which is cited in the sentence which follows.

Line 397: "The turbulent diffusion step in the model is also prone to creating negative mass, but to a lesser degree." Can the authors please provide a reference for this? Standard second-order diffusion does not create negative mass (see e.g., Fig. 2a in Xue (2000): High-Order Monotonic Numerical Diffusion and Smoothing).

We used the sixth-order diffusion scheme without the monotonic option as this is not conservative, which as you can see in Fig. 2c in Xue (2000) can create negative values. Relevant discussions were added in Section 2 first paragraph.

Lines 463 and 473: use singular "Table 6" Done (new Table 7).

Line 465: "with model input emission rates (MIE)" Revised

Line 481: Why negligible estimated emission rates? The estimated emission rates for case 2 were within 5% of MIE (except case CNRL0).

Revised to ", which resulted in negligible estimated emission rates based on flux calculations alone (i.e., not accounting for storage)".

Lines 485-490: Based on the results, I would be cautious to draw such a conclusion. Higher resolution does not always mean more accurate wind fields (see case 1, where d03 seems to be more accurate than d05). Where can I find the observation of (FC,H < 20%)?

Text was revised to "While both modelling setups replicated the same meteorological (advection) conditions, the relatively coarser resolution of GEM-MACH simulations resulted in larger computational (grid) diffusion and consequently larger downwind dispersion of tracer amounts (and larger predicted horizontal mass flux). This in part demonstrates the benefits of employing super-resolution over high-resolution modelling. The WRF super-resolution simulations in this work were successful in closely replicating the observed weak advection conditions and GEM-MACH predicted FC,H<20% for the same period (Fathi et al., 2021), but at a higher spatio-temporal resolution. "

Line 502: "as indicated in Fig. 11" Revised

Line 524: Where are these modifications described in the manuscript? Is this related to the hard coding of emissions?

Yes, they are, see the revised text in lines 201-203 of the revised manuscript: "To simulate the emission of passive-tracers, the WRF dynamical-solver source code was modified following an approach similar to Blaylock's (as described in Blaylock 2017) used in Blaylock et al. (2017)."

Reference: Blaylock, B. K.: Tracer Plumes in WRF (last accessed: March 8, 2023), https://home.chpc.utah.edu/~u0553130/Brian_Blaylock/tracer.html, 2017.

Lines 536-538: This is quite speculative unless the authors provide more details on the numerical schemes.

The corresponding text in the Conclusion section was revised to "Our results suggest that the unbalanced creation of erroneous mass at sharp concentration gradients, such as at the vicinity of point sources, is intensified on an irregular grid (an artifact of model dispersion). However, more investigations with longer simulation times (beyond the scope of this

paper) are required to further investigate such effects. Small negative diffusive fluxes and the use of the positive-definite re-normalization scheme in our modelling setup prevented nonphysical effects (e.g., negative mass creation) for 10 out 11 emission sources (assigned on a regular grid). "

Lines 543-544: What happens if the tracer plume is vertically advected into layers with an irregular grid spacing? Then there is the same problem with losing mass.

That's correct, and we have discussed it in the last paragraph of Section 3 "Estimates were also partially affected by the changing vertical grid spacing for upper model layers, for tracer amounts mixing to higher altitudes (> 500 m). This is similar to mass loss for CNRL0 emissions, but to a much lesser extent."

Lines 552-553: See the previous comment.

Response to Anonymous Referee #2

**Comment on egusphere-2022-1125**

Anonymous Referee #2

Referee comment on "Passive Tracer Modelling at Super-Resolution with WRF-ARW to Assess Mass-Balance Schemes" by Sepehr Fathi et al., EGUsphere, https://doi.org/10.5194/egusphere-2022-1125-RC2, 2023

"Passive Tracer Modelling at Super-Resolution with WRF-ARW to Assess Mass-Balance Schemes" by Sepehr Fathi, Mark Gordon, and Yongsheng Chen

**Recommendation:** Major revisions

**General comments:**

RC2_01: This manuscript introduced a turbulence-resolving (or super-resolution) passive tracer modeling system and assessed its performance in terms of 1) meteorological parameters in the atmospheric boundary layer (ABL), 2) characteristics of small-scale passive tracer structures in the ABL, and 3) conservation of passive tracers. The super-resolution tracer modeling system was developed based on a one-way nesting capability of the WRF model. Specifically, a gradual downscaling from reanalysis scale (31.25 km) to an LES scale (50 m) was implemented in horizontal, and grid-refining in vertical was applied for the two innermost sub-kilometer grid-spacing domains. Meteorological evaluations were performed for three cases in comparison to station and aircraft observations of surface pressure, temperature, humidity, and wind. The first major concern I have is about the methodology used in meteorological evaluations, especially the inconsistency of the reference data (base case), i.e., the use of d03 results for evaluation of d04 results vs the use of d04 results for evaluation of d05 results etc. The characteristics of plume dispersion were then investigated focusing on differences caused by emission scenarios — e.g., emission source height, and emission type (point, line, and surface) — and by meteorological conditions — i.e., three different weather cases. My second major comment is that this manuscript lacks discussions on role of the ABL turbulent mixing in the plume characteristics, even though the results presented in the manuscript indicate that the ABL turbulence plays a significant role in determining the tracer dispersion that appears differently according to the emission source height and meteorological conditions.

AC2_01: We thank you for your constructive and detailed comments. We implemented your suggestions in revising our manuscript. Revisions in response to each comment are provided below.

**Major comments:**

RC2_02: 1. Revisions are needed in the methodology used in meteorological evaluations.

1.1. Wind speed, which is the most critical meteorological variable in accurate prediction of tracer dispersion modeling, shows large biases in comparison to observations. The authors suggested possible reasons of the large bias, including the NARR reanalysis data that have positive biases. To take account of the impacts of this reanalysis error carried over to the simulations, the authors used coarser resolution simulations to evaluate nested-domain simulations, instead of observations. I agree to the impact of reanalysis errors on the nested domains, but I think evaluation results in comparison to both observations and coarser-domain results need to be presented together, together with tables summarizing bias and RMSE scores, in the main text.

AC2_02: Thank you for your suggestion. A new table was added (the new Table 4) comparing domain d05 meteorological fields to WBEA observations.

Table 4. Evaluation of domain d05 simulations against d04 output fields at every 100 seconds at the geographical location of the CNRL oil sands facility (Lon=-111.738 and Lat=57.339). Note that positive/negative sings indicate over/under-estimates by d05 relative to d04.

| | | SLP (hPa) | 2m RH (%) | 2m T (°C) | 10m U (m/s) | 10m V (m/s) |
|---|---|---|---|---|---|---|
| | d04 mean | 1006.42 | 51.18 | 19.32 | 5.05 | 3.20 |
| Case 1 | rms error | 0.11 | 1.89 | 0.25 | 3.16 | 1.37 |
| | mean error | -0.04 | -1.20 | -0.17 | 3.11 | 1.14 |
| | d04 mean | 1013.44 | 52.59 | 20.30 | -0.766 | 0.799 |
| Case 2 | rms error | 0.56 | 7.59 | 1.84 | 0.59 | 2.20 |
| | mean error | 0.55 | 6.86 | -1.81 | -0.55 | -2.16 |
| | d04 mean | 1006.14 | 62.40 | 21.95 | 3.57 | 3.90 |
| Case 3 | rms error | 0.09 | 2.44 | 0.38 | 2.77 | 1.01 |
| | mean error | 0.06 | -2.24 | 0.36 | 2.73 | 0.75 |

RC2_03: 1.2. This study used percentage error as an evaluation metric. Most meteorological variables, except for wind speed, have large absolute values (e.g., pressure, RH, temperature, and wind direction), therefore using the percentage error as an evaluation metric could mislead about the

performance of the modeling results. I suggest adding a table that summarizes bias and RMSE scores of the meteorological variables in comparison to observations, similar to Table 4.

AC2_03: Text was revised in Section 3 to present model evaluation in variable units instead of percentage error. Also, a new table was added (see the previous comment).

RC2_04: 1.3. There are a number of places that plots and main texts are inconsistent. Wind-rose diagrams in Figure 3 show that winds are from north-east and east, while the main text mentions that the wind directions are from west and west south west (Line 265, Page 11).

AC2_04: Note that in the first version of the manuscript in Fig 3, winds were depicted as blown towards, rather than blown from. Hence the information in the Figure and the text are consistent: winds are from west and west southwest. In the revised version of the manuscript, Fig. 3 is revised to show winds as blown from.

RC2_05: 2. This manuscript lacks discussions on the role of turbulent mixing in determining plume characteristics, while the results presented in the manuscript indicate it is critical to understand the differences of the plume characteristics by emission source height and also between cases.

2.1. Plume characteristics from the emission scenario CNRL0 indicate the different role of turbulent mixing across the atmospheric boundary layer (ABL) top: i.e., within the ABL where turbulent mixing plays a dominant role in vertical structure of the ABL, including tracer concentration, vs. above the ABL where turbulent motions are suppressed by negative buoyance of the stably stratified inversion layer overlying the ABL.

Powered by TCPDF (www.tcpdf.org)

In Case 1, the plume dispersion resulting from the emission scenario CNRL0 shows very different behaviors (e.g., Figure 7) from other stack scenarios. The source height of the CNRL0 scenario is located at 483 m, which is around the ABL top where the vertical mixing by turbulence ceases due to the capping temperature inversion. The ABL height can be inferred from the potential temperature profile in Figure C1, which shows a stably stratified layer with small spatial variability (standard deviation) around 500 m. The tracers emitted from other sources are quickly (within a few minutes) mixed in vertical within the ABL from the surface to the ABL top, leading to a similar vertical structure at ~ 10 km downstream regardless of the source height (Figure 7). On the other hand, the vertical dispersion of tracers in the

CNRL0 scenario is confined to a smaller vertical domain, due to the relatively weaker turbulence mixing above the ABL top than within the ABL.

2.2. Differences in plume characteristics between Case 2 vs. other two cases can also be explained by the role of turbulence mixing. The meteorological profiles shown in Figure C2 indicate stronger turbulence activities in Case 2 than in other two cases, with the ABL top at around 1300~1500 m. In Case 2, I guess the plume dispersion of the CNRL0 would be very similar to other CNRL scenarios (though not shown nor mentioned in the manuscript), because all sources are located within the ABL in this case. Based on the meteorological profiles of Case 3 presented in Figure C3, I think Case 1 and Case 3 would show very similar results; the ABL top is located around 500 m in both cases, resulting in only the CNRL0 emission source being above the ABL top. More in-depth comparison of the plume characteristics between Case 1 and Case 2 could be made, based on the different role of turbulence mixing between the two scenarios. The manuscript did not provide any of these important points.

AC2_05: Thank you for your suggested discussion. Figures C1, C2, C3 were revised, a new figure for case 2 was added to the supplement, and the following text was added to the end of Section 3.3 "Average inversion heights $Z_i$ (inferred from potential temperature profiles) for the three cases are marked with dashed lines in Figs. C01, C02, and C03. $Z_i$ for cases 1 and 3 was between 300-400 m agl, placing the tracer sources (all except CNRL0) within the atmospheric boundary layer (ABL) where turbulent mixing plays a dominant role in modifying the vertical structure of the atmosphere including tracer concentrations. As a result tracer amounts released from these sources (at different heights) were mixed quickly in the vertical extent of the ABL within the 10 km downwind distance resulting in similar uniform vertical profiles (Fig. 7). For cases 1 and 3 the CNRL0 release height at 483 m agl was above $Z_i$, where turbulent mixing is suppressed by negatively buoyant atmosphere in the stably stratified inversion layer. Which confined the dispersion of CNRL0 tracer amounts within a smaller vertical extent and detached from the ground surface up to 10 km downwind distance (see Fig. 7). For case 2, $Z_i$ was between 1400-1500 m agl, placing all source including CNRL0 well within the ABL and resulting in similar vertical mixing for tracers released at different heights from ground surface up to 483 m agl (see Figure S9)."

[Figure]

Figure S9. Domain d05 west-east vertical cross-section for case 2 on 26 August 2013. Vertical cross-section of tracer plumes from stack/point emission sources are shown with release heights indicated for each source. Data shown is the tracer amount summed in the horizontal level and normalized to the maximum value for each source. The origin for south-north distance in km is at domain centre.

---

## Author Response (AR2)

**EGUSPHERE-2022-1125 | Development and technical paper**

Submitted on 20 Oct 2022

**Passive Tracer Modelling at Super-Resolution with WRF-ARW to Assess Mass-Balance Schemes**

Sepehr Fathi, Mark Gordon, and Yongsheng Chen, sepehr.fathi@ec.gc.ca

**Handling topical editor**: Jinkyu Hong, jhong@yonsei.ac.kr

**Author Responses to Referee Comments**
**Submitted on 01 Jun 2023: Anonymous referee #2**

**Color Code**
**Referee Comments in Grey**
**AC - Author Comments in Blue**
**(Sepehr Fathi, 13 June 2023)**

Referee #2:

"Passive Tracer Modelling at Super-Resolution with WRF-ARW to Assess Mass-Balance Schemes" by Sepehr Fathi, Mark Gordon, and Yongsheng Chen

Recommendation: Minor revisions

General comments:
I appreciate authors addressing my two major concerns. The manuscript has been improved from its initial submission, and I think it can be accepted for publication after minor revisions.

Minor comments:
1. Abstract and introduction
- P1, L1: Different model resolution criteria were used to define "super-resolution" throughout the manuscript, and please unify it. For example, 250 m was used at L1, 50 m at L14, and 100 m at L29.
AC_1: Revised
L14: ($\Delta x = 50m, \Delta z \simeq 10m, \Delta t \simeq 0.2s$)
L29: (e.g., $\Delta x < 100m, \Delta t < 1$)

- P1, L3: "top-down retrieval" => "top-down emission-rate retrieval"
AC_2: Done

- P1, L5: "top-down retrievals" => "top-down retrievals of emission rate"
AC_3: Done

- P1, L12: "sub-grid TKE" => "sub-grid turbulence" AC_4: Done

- P2, L36: "for inverse method analysis" of what?
AC_5: revised to "for inverse method analysis of emission rates."

2. Section 2.2. Model and Technical Setup
- P6, L131: Specify the version of the WRF model used in this study (e.g., V3.8, V4.0, etc.), and cite the WRF technical note for the version used.
AC_6: Done

- P6, L139: Specify the order of spatial discretization methods used.
AC_7: Mentioned in the following sentence.

- P8, L194: "dispersion" => "diffusion" AC_8: Done

- P9, L199: "K Option was set to" => "For vertical and horizontal diffusion by sub-grid turbulence, K Option was set to" AC_9: Done

- Figure 2: "decreasing size" => "decreasing domain size" AC_10: Done

3. Section 3.1. Model Sensitivity
The authors used terminologies "evaluation" and "error" for comparison between d04 and d05 (and also between d03 and d04) in Section 3.1 Model Sensitivity, but they should be replaced with "comparison" and "difference". This would better distinguish comparison between different model resolution results (Section 3.1) vs. evaluation of simulation results against observational data (Section 3.2). Detailed suggestions are listed below.

- P12, Table 4: "Evaluation of" => "Comparison of" AC_11: Done

- P12, L278: "evaluated the performance of" => "compared" AC_12: Done

- P12, L278: "The evaluations were made" => "The comparison was made" AC_13: Done

- P12, L280: "Root mean square (rms) error scores" => "Root mean square (rms) differences" AC_14: Done

- P12, L281 and L286: "rms errors" => "rms differences" AC_15: Done

4. Section 3.2. Meteorological Evaluation
- P14, L301, "by interpolating": Please, specify what interpolation method you used (e.g., linear interpolation).
AC_16: Revised, "linearly…"

- P16, L348, "Model wind fields ~ within on standard deviation for the three cases": This is not true for Case 1. It should be mentioned that "except for Case 1".
AC_17: Revised to "...within 1-2 standard deviation."

- P16, L349, "WBEA wind speeds are lower than both model and aircraft wind speeds": This is not true for Case 2. It should be mentioned that "except for Case 2".
AC_18: Revised to "WBEA wind speeds are on average lower than aircraft measured values for all cases, and less than model wind speeds for cases 1 and 3."

5. Section 3.3. Plume Characteristics
- P16, L363: "after start-up" => "after 1-hr spin-up" AC_19: Done
- P18, L367–L368, "covered less downwind range": I disagree to this. If you compare the east boundary of the model domain, surface emissions covered more downwind range compared to stack emissions.
AC_20: Note that tracer release points for different sources are at different distances to the domain eastern boundary. Text was added to further clarify this "… covered less downwind range **(downwind distance from the point of release)**".

- P20, L397–L413: There are a number of misinterpretations about the stability of the atmosphere in these paragraphs. I think discussions and analyses referring to the gradient Richardson number should be removed or rewritten to revise wrong statements in the current manuscript.

First of all, all three cases seem to be under thermally unstable conditions, from the characteristics of the plume dispersion and the mean temperature profiles that are well mixed below Zi. I assume case 2 is more purely thermally unstable with no and/or very weak wind shear, while other two cases are both thermally (e.g., by surface daytime heating) and mechanically (e.g., by vertical wind shear) unstable. The two paragraphs should be revised to correct that all three cases are under unstable conditions, not only Case 2 as stated wrongly in the current manuscript. Second, the gradient Richardson number computed using adjacent vertical grid points is not appropriate to diagnose the instability for Cases 1–3, i.e., in the atmospheric boundary layer (ABL) driven by daytime surface heating. One thing to consider regarding the gradient Richardson number in thermally-driven unstable ABL is that when there are surface heating and dry convection is generated by it, turbulent mixing can be very active throughout the ABL even though local temperature gradient and the gradient Ri are larger than zero; i.e., a strong turbulence mixing by large-scale convection happens even though dtheta/dz > 0 and gradient Ri > 0. In this case, using a gradient Ri can mislead the instability of the ABL, as wrongly interpreted in this manuscript. What is frequently used to diagnose instability of the ABL is the Richardson number computed using surface-layer variables (e.g., surface fluxes of momentum and heat), which can take account of the effects of surface heating and shear. Similar to the first point, the two paragraphs should be revised to remove the analysis using the gradient Richardson number and correctly mention that all three cases are under unstable conditions, not only Case 2.

Third, the critical Ri of 0.25 used in this study is suitable for shear-driven turbulence, and the critical Ri frequently used for thermally unstable condition is 0.0. Even with the critical Ri of 0.0, using the gradient Richardson number would lead to misinterpretation of the instability of the ABL, as I mentioned in the second point.

AC_21: Thank you for pointing out this correction in the manuscript. In fact all three cases were under unstable conditions, at different rates. The gradient Richardson number for all three cases was below the critical value for the bottom 1/3 to 1/2 of the ABL. The following revisions were made, and text added to clarify this:

- Old line 400, new 404 line: **"As a result of (thermal and dynamical) unstable conditions**, tracer plumes from emission sources mixed in the vertical up to 2000 m during the simulation time."

- Old line 407, new line 412: "Atmospheric conditions during this case were fairly **constant**…"

- Text was added to old line 409, new line 413: **"Note that for this case, similar to the other two cases, atmospheric conditions were unstable (both thermally and dynamically) within the bottom 1/3 to 1/2 of the ABL."**

- Conclusion line 568: "During case studies on 20 August and 2 September, atmospheric conditions **were less variable and the vertical wind shear was weak**, with higher wind speeds of about 5-15 m/s."

---

## Author Response (AR3)

**EGUSPHERE-2022-1125 | Development and technical paper**

Submitted on 20 Oct 2022

**Passive Tracer Modelling at Super-Resolution with WRF-ARW to Assess Mass-Balance Schemes**

Sepehr Fathi, Mark Gordon, and Yongsheng Chen, sepehr.fathi@ec.gc.ca

**Handling topical editor**: Jinkyu Hong, jhong@yonsei.ac.kr

***Color Code***
*Editor Comments in Grey (24 June 2023)*
*AC - Author Comments in Blue*
*(Sepehr Fathi, 4 July 2023)*

Author Responses to Editor Comments Submitted on 24 Jun 2023:

Public justification (visible to the public if the article is accepted and published)**:**

I ask revision of abstract. Abstract consists of two paragraphs. First paragraph is lengthy with sentences to emphasize the super-resolution modeling and second one summarizes this study with relatively short information. Please concise the sentences for the important aspects of this study and put more relevant information on the summary.

Abstract was revised as follows:

"Super-resolution atmospheric modelling can be used to interpret and optimize environmental observations during top-down emission rate retrieval campaigns (e.g., aircraft-based) by providing complementary data that closely correspond to real-world atmospheric pollution transport and dispersion conditions.  For this work, super-resolution model simulations with Large-Eddy-Simulation sub-grid scale parameterization were developed and implemented using WRF-ARW (Weather Research and Forecasting - Advanced Research WRF). We demonstrate a series of best practices for improved (realistic) modelling of atmospheric pollutant dispersion at super-resolutions. These include careful considerations for grid quality over complex terrain, sub-grid turbulence parameterization at the scale of large eddies and ensuring local and global tracer mass-conservation. The study objective was to resolve small dynamical processes inclusive of spatio-temporal scales of high-speed (e.g., 100 m/s) airborne measurements. This was achieved by downscaling of reanalysis data from 31.25 km to 50 m through multi-domain model nesting in the horizontal and grid-refining in the vertical. Further, WRF dynamical-solver source code was modified to simulate the release of passive-tracers within the finest resolution domain. Different meteorological case studies and several tracer source emission scenarios were considered. Model-generated fields were evaluated against observational data (surface monitoring network and aircraft campaign data) and also in terms of tracer mass-conservation. Results indicated agreement between modelled and observed values within 5 °C for temperature, 1-25% for relative humidity, and 1-2 standard deviations for wind fields. Model performance in terms of (global and local) tracer mass conservation was within 2% to 5% of model input emissions. We found that to ensure mass conservation within the modelling domain, tracers should be released on a regular resolution grid (vertical and horizontal). Further, using our super-resolution modelling products, we investigated emission rate estimations based on flux calculation and mass-balancing. Our results indicate that retrievals under weak advection conditions (horizontal wind speeds < 5 m/s) are not reliable due to weak correlation between the source emission rate and the downwind tracer mass flux. In this work we demonstrate the development of accurate super-resolution model simulations useful for planning/interpreting/optimizing top-down retrievals, and discuss favourable conditions (e.g., meteorological) for reliable mass-balance emission rate estimations."

---

## Author Response (AR4)

**EGUSPHERE-2022-1125 | Development and technical paper**

Submitted on 20 Oct 2022

**Passive-Tracer Modelling at Super-Resolution with WRF-ARW to Assess Mass-Balance Schemes**

Sepehr Fathi, Mark Gordon, and Yongsheng Chen, sepehr.fathi@ec.gc.ca

Handling topical editor: Jinkyu Hong, jhong@yonsei.ac.kr

**Editor Comments (24 Jul 2023):**
Topic editor decision: Publish subject to technical corrections.
by Jinkyu Hong
Public justification (visible to the public if the article is accepted and published):
The permanent links for code availability were not explicitly described in the manuscript. Please check it again.

**Author response (28 Jul 2023):** the permanent links for code and data were added to the main body of the manuscript at the following locations:

Section 2.2, 1st paragraph: "For this work, the Weather Research and Forecasting (WRF version 3.9, https://www2.mmm.ucar.edu/wrf/users/download/get_source.html, Skamarock et al., 2008) …"

Section 2.2, 4th paragraph: "The NARR data used for this study can be accessed at NOAAFathi (2022)."

Section 3.2, 1st paragraph: "The WBEA data used in this study can be accessed at WBEA-Fathi (2022)."

Section 3.2, final paragraph: "We also compared wind fields from domain d05 to aircraft observations during the JOSM 2013 campaign over the oil sands region for the same time periods as our model simulations (ECCC, 2013)."